# Parental relatedness through time revealed by runs of homozygosity in ancient DNA

Harald Ringbauer[1,2 ✉], John Novembre [2,3,4] & Matthias Steinrücken[2,3,4]

Parental relatedness of present-day humans varies substantially across the globe, but little is known about the past. Here we analyze ancient DNA, leveraging that parental relatedness leaves genomic traces in the form of runs of homozygosity. We present an approach to identify such runs in low-coverage ancient DNA data aided by haplotype information from a modern phased reference panel. Simulation and experiments show that this method robustly detects runs of homozygosity longer than 4 centimorgan for ancient individuals with at least 0.3 × coverage. Analyzing genomic data from 1,785 ancient humans who lived in the last 45,000 years, we detect low rates of first cousin or closer unions across most ancient populations. Moreover, we find a marked decay in background parental relatedness co-occurring with or shortly after the advent of sedentary agriculture. We observe this signal, likely linked to increasing local population sizes, across several geographic transects worldwide.

[1] Department of Archaeogenetics, Max Planck Institute for Evolutionary Anthropology, Leipzig, Germany. [2] Department of Human Genetics, University of Chicago, Chicago, IL, USA. [3] Department of Ecology and Evolution, University of Chicago, Chicago, IL, USA. [4]These authors contributed equally: John Novembre, Matthias Steinrücken ✉email: harald_ringbauer@eva.mpg.de

An individual's parents can be related to each other to varying degrees. For present-day humans, much intriguing geographic variation in parental relatedness has been observed. On one end of the spectrum, globally more than 700 million living humans are the offspring of second cousins or closer relatives. In some regions, the rate of such unions reaches 20–60%[1]. Parents can also be more distantly related to each other, often via many deeper connections in their pedigree, as a common consequence of small population sizes[2–4], or as a consequence of founder effects in tight-knit groups[5,6]. At the other end of the spectrum, in large populations where cousin marriages are not common, many parents have no recent connections in their pedigree at all[2]. Going back in time, sporadic matings of close kin are documented in royal families of Europe, ancient Egypt, Inca, and pre-contact Hawaii[7,8], but little is known about broader patterns of past parental relatedness, because archeological evidence alone is typically not informative about mating preferences, especially for prehistoric societies.

The genetic sequence of an individual contains information about the relatedness of their two parents since co-inheritance of identical haplotypes results in stretches of DNA that lack genetic variation (Fig. 1A, often termed runs of homozygosity[5], though also known by other terms, such as segments with homozygosity by descent (HBD, Supplementary Note 4). The more recent the genealogical relationship of the two parents, the more frequent and longer the resulting ROH tends to be[2]. ROH can be identified in genome-wide data[9,10], and this signal has been analyzed for a wide range of purposes in medical, conservation, and population genetics[2].

Recently, ROH has been identified in ancient DNA (aDNA)[11–18], that is, genetic material extracted from ancient human remains. This advance is especially promising, as large datasets of aDNA have been generated in the last decade[19].

However, major challenges persist, coverage for aDNA is often around or less than 1× per site (see Fig. S19), and contamination and DNA degradation introduce genotyping errors[20]. As a consequence, ROH detection is currently only possible for ancient individuals of exceptional high coverage. Recent methodological advances enable identifying ROH in data with at least 5× coverage[21], but this threshold precludes analysis on all but a small fraction of the currently available aDNA record.

Here, we present an approach to detect ROH that can identify ROH longer than 4 centimorgan (cM) in individuals with coverage as low as 0.3×. It is designed to perform well for a common type of human aDNA data: Pseudo-haploid genotypes (Fig. 1B), which consist of a single allele call for each diploid site. Such data do not convey homozygote versus heterozygote genotype states directly; however, as we show, one can extract ROH from such data by leveraging haplotype information from a phased reference panel.

Using this method, we analyze 1785 ancient individuals from the last 45,000 years, a substantial fraction of the published global human aDNA record. We focused on quantifying two domains of parental relatedness: (1) Close-kin unions, measured by the sum of all ROH > 20 cM, denoted as sROH > 20; and (2) background relatedness as measured by the sROH 4–8 cM. First, we find that matings among first-cousins or closer relatives, though widely practiced today in numerous societies, are generally infrequently observed in aDNA data. Second, we observe decreasing levels of short ROH across many regions coinciding with or shortly after the local Neolithic transition from foraging to agricultural subsistence strategies. This genomic evidence of reduced background relatedness supports and refines long-held evidence of the Neolithic transition involving a major demographic shift towards increased local population sizes.

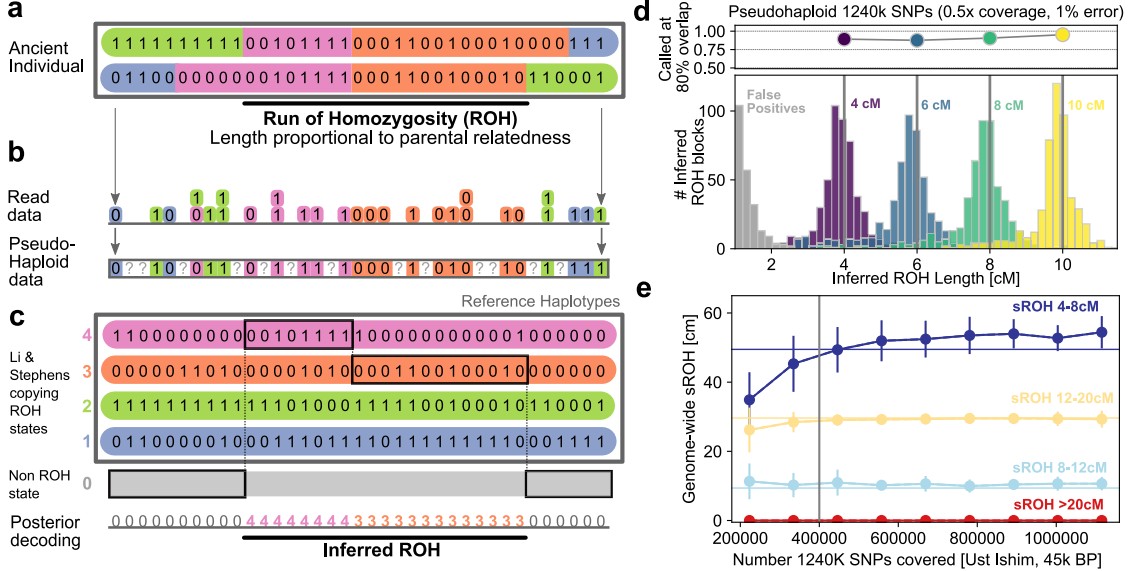

**Fig. 1 Detecting runs of homozygosity using a reference panel. a** Illustration of genotype data for a diploid individual. **b** Mapping sequencing reads to a biallelic SNP produces counts of reads for each allele, from which in turn pseudo-haplotype genotypes, i.e., single reads per site, are sampled. **c** Schematic of Method. A target individual's genotype data is modeled as mosaic copied from haplotypes in a reference panel (ROH states, colored) and one additional background state (non-ROH, gray). **d** We applied our method to simulated data with known ROH copied in (see Supplementary Note 1.7 for details). We copied 500 ROH of exactly 4, 6, 8, and 10 cM into 100 artificial chromosomes, and depict histograms of inferred ROH lengths (in color) as well as false positives (in gray) after downsampling and adding errors (0.5× of all 1240K SNPs, with 1% error added). **e** Down-sampling experiment of a high coverage ancient individual who lived 45,000 years ago (Ust Ishim man) using a modern reference panel (1000 Genomes dataset). We down-sampled pseudohaploid data at the 1240K SNPs. For each target coverage, we ran 100 independent replicates and depict the mean and standard deviation of the inferred ROH in four length bins (4–8, 8–12, 12–20, and >20 cM). The horizontal lines indicate high confidence estimates using diploid genotype calls from all available data.

## Results

**Detecting ROH using a haplotype reference panel.** Our approach to detect ROH employs a phased reference panel to leverage haplotype data (described in Supplementary Note 1). Briefly, our method utilizes the fact that sequencing reads in a region of ROH are effectively sampled from a single haplotype only because the maternal and paternal haplotypes of the diploid individual are identical. In contrast, outside an ROH, two distinct haplotypes are carried by the target individual, and thus, the sequencing reads to originate from both. As a result, modeling sequencing reads as a mosaic of long stretches copied from single reference haplotypes works substantially better within ROH regions (see Fig. 1). To utilize this signal, we developed a Hidden Markov Model (HMM) with hidden copying states, one for each reference haplotype, to model copying long stretches from the panel [similar to the copying model of ref. [22]], and an additional single non-ROH state as in ref. [10]. We implemented this algorithm in the software package hapROH, available at https://pypi.org/project/hapROH. The default parameters of the current implementation are tuned for pseudo-haploid genotype calls from a widely used capture technology that targets ca. 1.24 million SNPs [hereafter the "1240K" SNP panel[23]] when using a reference panel of 5008 haplotypes of present-day human genetic variation [1000 Genomes[24]].

**Validating the ROH inference.** We tested the method in four scenarios: (1) Spiking ROH segments of various lengths (4–10 cM) into data (Supplementary Note 2.1), (2) down-sampling high-coverage ancient individuals (Supplementary Note 2.3), (3) down-sampling present-day individuals (Supplementary Note 2.4), and (4) testing different divergence times between the reference panel and the target individual (Supplementary Note 2.2).

For the spike-in experiments, we observe that the power to detect at least 80% of an inserted ROH block was above 85% in all simulated scenarios (Fig. 1D, Supplementary Note 2.1). Bias in the estimated length of the longest overlapping inferred ROH was consistently below 0.5 cM, and we observed no false positives for ROH > 4 cM. In the down-sampling experiments, we tested the ability to recover the sum of ROH segments falling into four length ranges (4–8, 8–12, 12–20, and >20 cM). When down-sampling the oldest modern human genome sequenced to high depth [Ust-Ishim, 45,000 years old[25]], the method produces little bias in estimating the respective sROH statistics from pseudo-haploid data with as few as 400,000 of the 1240K sites covered (see Fig. 1E, Supplementary Note 2.3). In the experiments where we down-sampled 599 present-day individuals from a global sample (Supplementary Note 2.4), the ROH inference from pseudo-haploid data performs generally well ($sROH_{[4,8]}$: $r = 0.925$ between diploid ROH calls and pseudo-haploid data, $sROH_{>20}$: $r = 0.988$, Fig. S10), except for sub-Saharan forager populations. When assessing the impact of different divergence times between the test population and the haplotype reference panel in individuals with a simulated coverage of 1×, we find that using a European-only reference panel, the method can detect ROH in low coverage test individuals from East Asia and South America but showed much less power for test individuals from West Africa (see Fig. S6).

Together, our tests show that the method can infer ROH segments longer than 4 cM for individuals with more than 400,000 of the 1240K sites covered at least once while tolerating sequencing error rates up to 3%. In addition, our experiments demonstrate that the method can analyze target individuals from populations that diverged from the reference panel up to several ten thousand years ago. Therefore, all present-day and ancient humans that share the out of Africa bottleneck [20,000–40,000 BP[26]] fall into the range of applicability of our method when using the 1240K marker set and the full 1000 Genomes dataset as haplotype reference panel.

**Application to aDNA data.** We then applied our method to a large publicly available dataset of aDNA data (Allen Ancient DNA Resource v42.4, released on March 1, 2020) using the 1000 Genomes dataset as haplotype reference panel (see Section "Methods"). Only 134 of the 3723 individuals in this dataset have average coverage > 5× (Fig. S19), a typical minimum coverage requirement for previous ROH methods[21]. Using the method described here with its threshold of 400,000 of the 1240K sites covered at least once allowed us to analyze a much larger fraction of this dataset (1833 of the 3723 individuals). We also integrated a dataset of modern individuals genotyped at the Human Origins SNPs [HO[27]], which are a subset of the 1240K SNPs. After quality control and filtering (see Section "Methods"), we arrived at a dataset of 1785 ancient and 1941 present-day unique individuals. Within this dataset, we inferred ROH longer than 4 cM using all available 1240K pseudo-haploid data in all ancient individuals and using diploid data for the HO SNPs in all modern individuals. After confirming that ROH calls on pseudo-haploid and diploid data in modern individuals correlate closely (Pearson correlation coefficient $r = 0.925$–0.988, Fig. S8), we analyzed the inferred ROH in ancient and modern individuals jointly.

**Low abundance of long ROH in ancient humans.** We first identified individuals with $sROH_{>20}$ greater than 50 cM. We chose this threshold based on calculations Supplementary Note 4 and simulations Supplementary Note 5, which show that in large populations, 88% and 20% of the offspring of first and second cousins, respectively, have $sROH_{>20} > 50$ cM, but less than 1% of offspring of third or more distant cousins do. The 50 cM threshold for $sROH_{>20}$ can also be surpassed in very small isolated populations, specifically, 34% of individuals in populations of size 250 and 8% for size 500 (Fig. S15). Hereafter we refer to this as the "long ROH" threshold, and individuals crossing it as having "long ROH".

Overall, we find that only 54 out of the 1785 ancient individuals (3.0%, CI: 2.3–3.9%) have $sROH_{>20}$ above 50 cM. Generally, these individuals with long ROH do not concentrate in any particular region or time period (Figs. 2B and 3). The only archeological cluster (defined in annotations from the source dataset, modified for readability) with more than two individuals is "Iron Age Republican Rome", where 3 of 11 samples (reported in ref. [28]) fall above the long ROH threshold. In the Pontic-Caspian Steppe region, 3 of 54 individuals who lived between 2600 and 1500 BP (5.6%, CI: 1.2–15.4%) exceed the threshold (Fig. 2F), but this signal is not significantly different from the rate in the full dataset. Three individuals with long ROH appear in the late pre-contact Andes region (Fig. 2D), and a follow-up study describes this signal with a larger sample size[29]. Notably, 11 of the 54 individuals with long ROH are located on islands: Ordered by time and using the cluster annotations from the publicly available dataset (modified for readability) these are: "Sardinia Early Copper Age" (1 of 1, Fig. S20), "Sweden Megalithic" (1 of 5, all on Gotland), "England Neolithic" (1 of 16), "Chilean Western Archipelago" (1 of 3), "England C-EBA" (2 of 14, Fig. 2), "Russia Bolshoy" (2 of 6), "Vanuatu 1100 BP" (1 of 3), "Argentina Tierra del Fuego" (1 of 1), and "Indian Great Andaman" (1 of 1).

The highest value of $sROH_{>20}$ across the whole dataset (including present-day individuals) is found in a 6000-year-old Levantine Copper Age individual [I1178[30]] with 545 cM $sROH_{>20}$. The other 8 individuals tested from the same burial

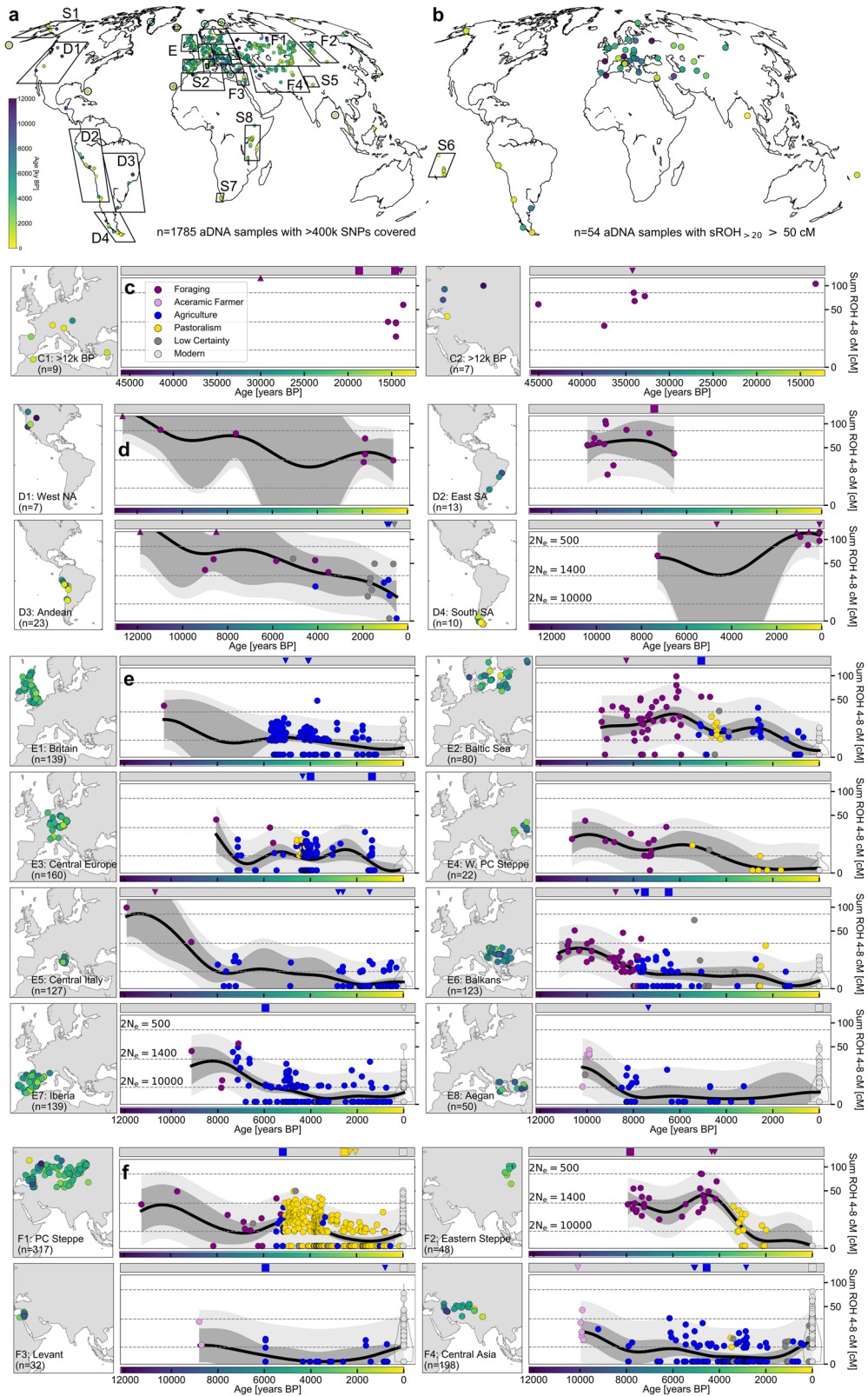

site [Peqi'in Cave, Israel Chalcolithic 6000 BP[30]] had sROH$_{>20}$ values of 0, and very little ROH overall (sROH$_{>4}$ < 30 cM). The sum and length distribution of ROH suggest the parents of individual I1178 were first-degree relatives (Fig. 4), i.e., parent-offspring or full siblings whose offspring will have a quarter of

their genome in ROH. We note that the burial context of this male individual was not reported to be exceptional.

The rate of long ROH is substantially higher in the present-day Human Origins dataset; we inferred that 176 of 1941 modern individuals (9.1%, CI: 7.8–10.4%) have long ROH. In contrast to

**Fig. 2 Time transects of major regions. a** Global distribution of ancient individuals screened for ROH. **b** Global distribution of ancient individuals with long ROH. **c–f** We plot sROH$_{[4,8]}$ for individuals (represented as circles) within several geographic transects (defined in Table S1). Mean estimates were calculated from a Gaussian Process (GP) model (solid black line, see Section "Methods"), as well as 95% empirical confidence intervals for both individuals (light gray) and for the estimated mean (dark gray). Note the square root scale (chosen for GP modeling, see Section "Methods"). Individuals with values larger than the upper y-axis limit are indicated on top of the panels (upward triangles). Horizontal dashed lines depict expectations for sROH$_{[4,8]}$ for panmictic population sizes (see formulas in Supplementary Note 4). In the gray bar at the top of each panel, we indicate individuals with sROH$_{>20}$ more than 100 (squares) and 50 cM (downward triangles), which are plausibly offspring of close kin (held out when fitting the GP). Where available, we show ROH in present-day individuals (light-gray points for each individual, violin plot for density estimate).

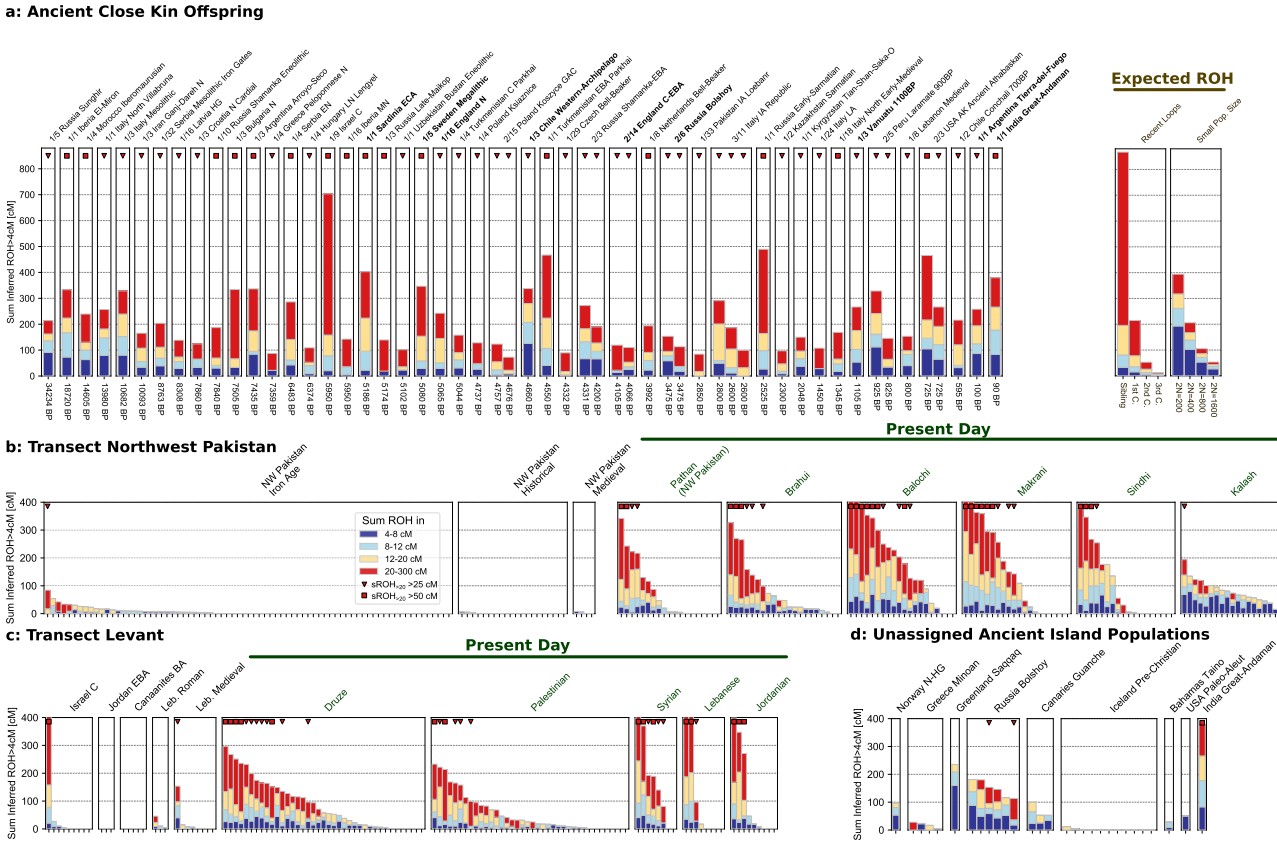

**Fig. 3 Individual ROH in a subset of ancient and present-day populations.** Each individual is represented by stacked vertical bars, where the length of each bar is determined by the ROH of this individual falling into four length classes (4–8, 8–12, 12–20, and >20 cM, color-coded). **a** All 54 ancient individuals (out of 1785) with at least 50 cM sROH$_{>20}$ (x/n indicates a number of individuals x exceeding the threshold in a cluster of size n defined by archeological label). The labels of the 11 individuals from island populations are highlighted in bold. We also show a legend (top right) of expected ROH for offspring of close kin or in small populations, based on analytical calculations (Supplementary Note 4). For simulations exhibiting individual variation around the mean see Supplementary Note 4 and Supplementary Note 5. **b**, **c** Time transects for regions covering present-day Pakistan and Levant, respectively. Modern individuals are indicated by the green horizontal bars. **d** Ancient individuals from island populations not assigned to geographic regions (circles in Fig. 2a).

ancient data, several geographic clusters of long ROH are found, mainly in present-day Near East, North Africa, Central/South Asia, and South America (Supplementary Data 1). This signal was described previously [reviewed in ref. [2]] and mirrors the estimated prevalence of cousin marriages[1].

In two regions where long ROH are common in the present-day data (Fig. 3) our ancient data contains several ancient individuals, which allowed us to analyze time transects. In the Levant, all five present-day annotated groups in our study (Druze, Palestinian, Syrian, Lebanese, Jordanian) have a high fraction of individuals above the long ROH threshold (30 out of 102 in total, see Fig. 3C). In the ancient sample of this region, only 2 out of 28 analyzed Levant individuals from the Copper Age (n = 9), Bronze Age (n = 8), Roman times (n = 3) to the Middle Age (n = 8) fall

above this threshold: the first is the Israel Chalcolithic individual with the highest sROH$_{>20}$ in our dataset (see above) (Fig. 4); the second is a male individual (SI-38) excavated from a mass burial in South Lebanon connected to a Medieval Crusader battle, who was found to have local ancestry[31]. The second region for which we could analyze a time transect is the region of present-day Pakistan. In five out of six modern annotated groups in the dataset (Pathan, Brahui, Makrani, Balochi, and Sindhi), many individuals have long ROH (33 out of 98 individuals with sROH$_{>20}$ above 50 cM). In the sixth group, from the Kalash, an isolated valley population, only 1 individual out of 18 exceeds this threshold, despite elevated levels of background ROH being observed (Fig. 3B). In contrast, in the ancient individuals[32] [from present-day Northwestern Pakistan], we infer that only 1

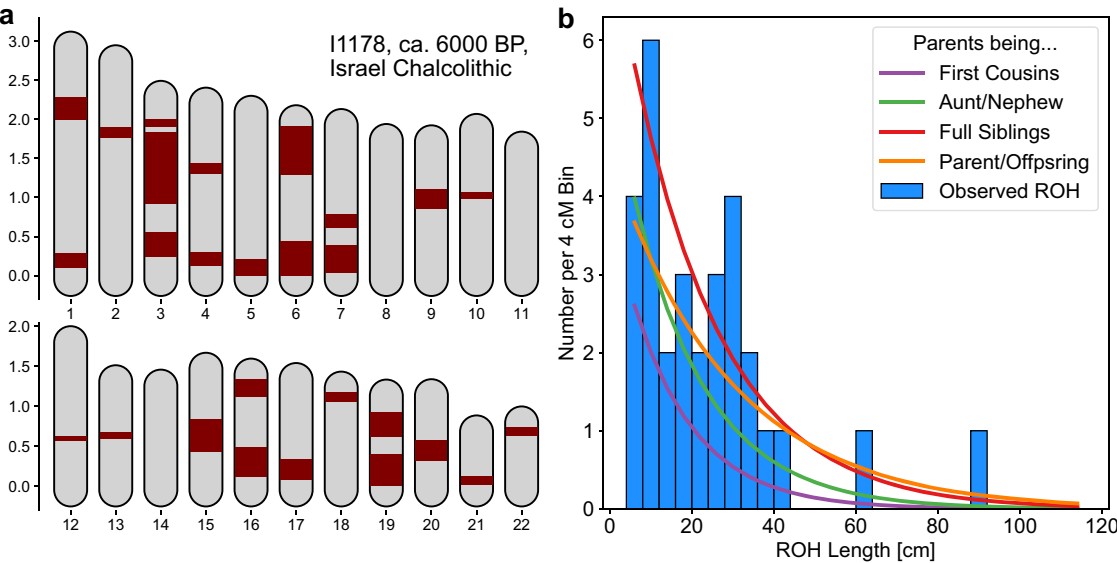

**Fig. 4 ROH in a 6000-year-old individual.** We show ROH of the individual with the highest sum of inferred ROH among all our samples, I1178, a male individual context-dated to ca. 4500–3900 BCE, reported in ref. [30]. We inferred 703.2 cM of his genome in ROH longer than 4 cM, with the longest ROH spanning 91.1 cM. **a** We marked the position of these ROH on the 22 autosomes (maroon), with map length annotated in Morgan. **b** We depict a histogram of the ROH lengths, together with expected densities of ROH for certain degrees of parental relationships, calculated as described in Supplementary Note 4.

individual out of 75 from the Iron Age (3200–2700 BP) has $sROH_{>20}$ above the threshold and that none of the 20 individuals from the Historical Period (2600–1900 BP) and none of the 4 individuals from the Middle Period (900–400 BP) surpass the long ROH threshold (Fig. 3B).

**Human background relatedness decreased over time.** Shorter ROH segments measured by $sROH_{[4,8]}$ accumulate from parental lineages coalescing on average 10–30 generations ago (Fig. S14); thus, their abundance reflects the size of the ancestral mating pool (background relatedness) over approximately the previous half millennium [assuming 30 years per human generation[33]]. Because ancestry often spreads out geographically back in time, the probability of recent coalescence and ROH decreases not only with increasing local population size but also increasing parent-offspring dispersal[34,35]. Assuming that individual mobility is comparable between groups, $sROH_{[4,8]}$ proxies for local population size[36]. We plotted the values of $sROH_{[4,8]}$ in time transects for 24 major geographic regions that cover 1763 of the 1785 ancient individuals (16 regions shown in Fig. 2, 8 regions in Fig. S20, 29 additional individuals from islands are shown in Fig. 3D, and the remaining 22 individuals are reported in Supplementary Data 1). In addition, we tested whether $sROH_{[4,8]}$ differs between subsistence strategies (annotated for most ancient individuals, see Section "Methods") in certain regions (PERMANOVA used for Table 1 and p-values in the text below).

We find that $sROH_{[4,8]}$ is highest among the most ancient individuals in the dataset and then generally decreases going forward in time. Each of 43 ancient individuals in the global sample dated to before 10,000 BP was inferred to have $sROH_{[4,8]} > 0$, with a median value of 54.5 (39 individuals shown in Fig. 2). We then observe a substantial decline in $sROH_{[4,8]}$ coinciding with the Neolithic transition to sedentary, agricultural lifestyles (Fig. 2). In Western Eurasia, we contrasted individuals from forager cultures to those from early farming cultures (i.e., farming cultures within the first 2000 years after the first annotated "Agriculture" individual per region). We found that $sROH_{[4,8]}$ decreases substantially in all 8 regional transects which

contain both annotated foragers and farmers (p-value < 0.05 in 7 of these 8 transects), and median $sROH_{[4,8]}$ values drop from 13 to 66 cM per foraging group to 0–9 cM per early farming group (Tab. 1). In the Andes, where agriculture gradually increased in intensity starting around 5000 BP in a heterogeneous process lasting thousands of years[37], the median $sROH_{[4,8]}$ decreases from 55.4 for foragers to 17.9 for agriculturalists ($p = 1.2 \times 10^{-2}$, Table 1).

Detailed inspection of the transitions from foraging to farming reveals interesting finer-scale dynamics. First, for the earliest western Eurasian farmers not using ceramics yet, who lived ~10,000 years ago and predate the Neolithic expansions into Europe, we still observe elevated rates of short ROH, with a median $sROH_{[4,8]}$ of 36.7, 16.4, and 15.2 in Aegean, Levant, and Central Asian aceramic farmers, respectively, which is comparable to values observed for western Eurasian foragers ($sROH_{[4,8]}$ ranging from 13 to 66 cM, Table 1). In all three regions there is a subsequent marked drop to ceramic early farmers, with median $sROH_{[4,8]}$ decreasing substantially to 0, 0, and 4.8, respectively ($p = 6.0 \times 10^{-5}$, $3.8 \times 10^{-2}$, and $5.4 \times 10^{-2}$, Table 1).

Furthermore, one ceramic early farming group in our sample stands out: Individuals annotated in the original dataset as Iberian Early Neolithic [7400–7000 BP[38]] have median $sROH_{[4,8]}$ of 32.8 cM, which is substantially higher than in other early Eurasian farmers (median $sROH_{[4,8]}$: 0–8.7 cM, Table 1). However, in Iberian Middle Neolithic farmers (6800–4600 BP) ROH decreases (median $sROH_{[4,8]} = 0$, $p = 1.0 \times 10^{-5}$, Table 1) and becomes typical of other early European farmers. As the early Iberian individuals have exceptionally high early farmer ancestry [>90%[38]], this signal cannot be explained by forager (hunter-gatherer) ancestry. However, archeological evidence of a rapid maritime spread (Cardial Ware expansion) within a few hundred years around 7500 BP[39] provides one plausible explanation of this increased abundance of short ROH in the Early Neolithic, as a rapid spread could have caused an initial bottleneck. Moreover, an initially small population of early farmers would explain why forager admixture substantially increased in Middle Neolithic Iberians and remained one of the highest of European Neolithic populations [~25%[38]].

**Table 1 Comparison of statistics for sROH$_{[4,8]}$ for pairs of groups.**

| Group A | Group B | $n_A$ | $n_B$ | sROH$_{[4,8]}$ A 25% | 50% | 75% | sROH$_{[4,8]}$ B 25% | 50% | 75% | p-Value |
|---|---|---|---|---|---|---|---|---|---|---|
| FG >10k BP | FG 8–10k BP | 35 | 53 | 29.2 | 51.0 | 81.6 | 9.9 | 20.8 | 39.2 | **3.5 × 10$^{-4}$** |
| **Foraging** | **Early Farmer** | | | | | | | | | |
| All Eurasian | | 111 | 223 | 7.2 | 18.7 | 37.5 | 0.0 | 4.6 | 9.4 | **1.0 × 10$^{-5}$** |
| Aegan | | 1 | 27 | 30.9 | 30.9 | 30.9 | 0.0 | 0.0 | 5.0 | **3.6 × 10$^{-2}$** |
| Balkans | | 38 | 35 | 5.6 | 13.6 | 21.0 | 0.0 | 4.2 | 8.5 | **1.0 × 10$^{-5}$** |
| Baltic Sea | | 41 | 7 | 9.3 | 22.2 | 38.8 | 6.9 | 8.7 | 12.7 | **3.5 × 10$^{-2}$** |
| Britain | | 1 | 90 | 39.8 | 39.8 | 39.8 | 4.2 | 5.6 | 9.4 | **1.1 × 10$^{-2}$** |
| Central Europe | | 4 | 17 | 26.6 | 36.9 | 46.7 | 0.0 | 0.0 | 4.5 | **3.6 × 10$^{-4}$** |
| Iberia | | 4 | 23 | 7.5 | 25.6 | 45.9 | 0.0 | 5.1 | 26.5 | 1.3 × 10$^{-1}$ |
| Central Italy | | 2 | 11 | 49.2 | 65.8 | 82.4 | 0.0 | 4.4 | 10.8 | **1.3 × 10$^{-2}$** |
| Steppe | | 20 | 13 | 5.6 | 17.3 | 51.7 | 0.0 | 4.3 | 9.5 | **9.9 × 10$^{-3}$** |
| Andean | | 7 | 5 | 46.8 | 55.4 | 95.2 | 9.6 | 17.9 | 22.1 | **1.2 × 10$^{-2}$** |
| Farmers > 5k BP | Steppe-PA 5.2-3k BP | 286 | 188 | 0.0 | 4.2 | 9.2 | 4.2 | 10.9 | 17.1 | **1.0 × 10$^{-5}$** |
| Central Asian > 3k BP | Steppe-PA 5.2-3k BP | 84 | 188 | 0.0 | 0.0 | 6.2 | 4.2 | 10.9 | 17.1 | **1.0 × 10$^{-5}$** |
| **Foraging** | **Pastoralism** | | | | | | | | | |
| Western PC Steppe | | 13 | 8 | 4.5 | 14.2 | 17.3 | 0.0 | 0.0 | 1.0 | **6.9 × 10$^{-3}$** |
| Eastern Steppe | | 28 | 18 | 24.3 | 32.5 | 49.9 | 0.0 | 4.7 | 12.4 | **1.0 × 10$^{-5}$** |
| **Aceramic Farmer** | **Early Ceramic Farmer** | | | | | | | | | |
| Aegan | | 5 | 27 | 36.2 | 37.2 | 38.0 | 0.0 | 0.0 | 5.0 | **6.0 × 10$^{-5}$** |
| Levant | | 2 | 11 | 11.0 | 16.4 | 21.8 | 0.0 | 0.0 | 2.0 | **3.8 × 10$^{-2}$** |
| Central Asia | | 5 | 5 | 11.4 | 15.2 | 25.8 | 0.0 | 4.8 | 7.0 | 5.4 × 10$^{-2}$ |
| Iberia-EN | Iberia-MN | 7 | 15 | 23.7 | 32.8 | 37.3 | 0.0 | 0.0 | 4.7 | **1.0 × 10$^{-5}$** |
| Americas | other < 13k BP | 57 | 1658 | 28.0 | 56.3 | 85.8 | 0.0 | 4.2 | 10.5 | **1.0 × 10$^{-5}$** |

For each of the two groups (A and B), we calculate the sROH$_{[4,8]}$ statistic with individuals having sROH$_{>20}$ > 50 cM removed. We report the total sample size per group ($n_A$ and $n_B$), quantiles of the sROH$_{[4,8]}$ statistic (25%, 50%, and 75%), and a two-sided p-value for the pairwise comparison (PERMANOVA with 99,999 Permutations, see Section "Methods", the minimal attainable p-value is 1.0 × 10$^{-5}$). All p-values < 0.05 are marked in bold (no multiple testing correction was applied). Early Farmer groups are defined as 2000 years after the first individual with an "Agriculture" annotation per region (Fig. 2). Other population abbreviations are FG foraging, PA pastoralism, EN early neolithic, MN middle neolithic.

In the ancient Americas, elevated sROH$_{[4,8]}$ values evidence sustained high levels of background relatedness. This signal is found across all American regions: Western North America (West NA) (Fig. 2D1, median sROH$_{[4,8]}$ = 67.1 cM excluding long ROH individuals, $n = 7$), Eastern South America (East SA) (Fig. 2D2, 59.0 cM, $n = 13$), Andean (Fig. 2D3, 31.5 cM, $n = 20$), Southern SA (Fig. 2D4, 112.5 cM, $n = 8$) and Beringia (Fig. S20, 52.4 cM, $n = 9$). This abundance of ROH (overall median sROH$_{[4,8]}$ = 56.3) is higher than the rest of the global sample in the same broad time period (<13k years ago, median 4.2 cM, $p < 10^{-5}$, Table 1). Since sROH$_{[4,8]}$ is driven by co-ancestry within the last few dozen generations (Supplementary Note 4), this elevated sROH$_{[4,8]}$ cM cannot be explained by bottlenecks during early migrations into the Americas, but one needs to invoke more recent, sustained small effective population sizes. Overall we observe little temporal variation (Fig. 2), with one exception in the dataset being Andean populations around the time of the shift to agriculture (see above; also note the dataset does not include individuals from other early centers of agriculture in the Americas, e.g., Central Mexico, eastern North America).

Another observation of elevated ROH on a large geographical scale is found in the Eurasian Steppe, where early pastoralist groups all have substantial amounts of sROH$_{[4,8]}$ (Steppe-PA 5.2-3k BP, median sROH$_{[4,8]}$ = 10.9, Table 1), including the Yamnaya (median 17.5 cM, $n = 17$), Afanasievo (18.1 cM, $n = 22$), Sintashta (5.7 cM, $n = 21$), Okunevo (24.5 cM, $n = 12$) and Srubnaya (4.8 cM, $n = 19$). These sROH$_{[4,8]}$ levels are significantly higher than in Western Eurasian farmer populations before 5000 BP (median 4.2 cM, $p = 1.0 \times 10^{-5}$, Table 1), and, notably, also significantly higher than their southern contemporaneous neighbors, sedentary farmers from Central Asia (median 0, $p = 1.0 \times 10^{-5}$, Table 1). In samples from the Western Pontic-Caspian Steppe (present-day Ukraine and Moldavia), at the transition from foragers to pastoralists, we observe a substantial decrease of sROH$_{[4,8]}$ from median 14.2 to 0 ($p = 6.9 \times 10^{-3}$, Table 1). Similarly in the Eastern Steppe (around Lake Baikal and present-day Mongolia), a shift from foragers to pastoralism coincides with a significant reduction in sROH$_{[4,8]}$ (median 32.5–4.7, $p = 1.0 \times 10^{-5}$). We note that in both the Western and Eastern Steppe many of the pastoralists in our sample date to 3000–2000 BP (Scythian and Xiongnu, respectively), substantially later than the early pastoralists mentioned above.

## Discussion

We developed a method for measuring ROH in low coverage ancient DNA. Our algorithm follows a long line of previous work utilizing HMMs to infer such segments[10,40–42]. A key methodological advantage here is to use hidden states that, within an ROH segment, copy from a reference panel of haplotypes to take advantage of haplotype information. This tool enabled us to screen aDNA data from 1785 individuals for ROH, an order of magnitude more ancient individuals than hitherto amenable for such analysis. We generated evidence for two key aspects of the human past: Identifying long ROH (>20 cM) provided insight into the past prevalence of close kin unions such as cousin matings, whereas short ROH (4–8 cM) revealed changing patterns of past background relatedness that reflect local population sizes.

We found that only 1 out of 1785 ancient individuals have long ROH typical for the offspring of first-degree relatives (e.g., brother–sister or parent–offspring). Historically, matings of first-degree relatives are only documented in royal families of ancient Egypt, Inca, and pre-contact Hawaii, where they were sporadic occurrences[7]. The only other example of an offspring of first-degree relatives found using aDNA to date is the recently

reported case from an elite grave in Neolithic Ireland[18]. Our findings are in agreement that first-degree unions were generally rare in the human past.

Further, we find that only 54 out of 1785 ancient individuals (3.0%, CI: 2.3–3.9%) have long ROH typical for the offspring of first cousins (88%) and less commonly observed for second cousins (20%). Such long ROH can also arise as a consequence of small mating pools (e.g., 8% in randomly mating populations of size 500, which may explain the long ROH we observed on certain island populations). Therefore, the rate of long ROH is an upper bound for the rate of first-cousin unions. On the other hand, because of incomplete power, some long ROH may be missed in our empirical analysis; however, even if the method would fail to detect half of all ROH > 20 cM, well below the power that we observed in our simulations, we would still detect 60% of first cousins (see Table S5). We conclude that in our ancient sample substantially less than 10% of all parental unions occurred on the level of first cousins.

In two specific regions with high levels of long ROH in the present-day[2], the dataset contained a sufficient number of ancient individuals to allow analyzing time transects. For both transects (the Levant and present-day Northwest Pakistan), we observe a substantial shift in the levels of long ROH. In contrast to the high abundance of long ROH typical of close kin unions in the present-day individuals, long ROH was uncommon in the ancient individuals, including up to the Middle Ages. Additional data from these regions and others with high levels of long ROH today, such as North Africa as well as Central, South, and West Asia[2], will help resolve with more precision the origin and spread of these well-studied kinship-based mating systems[43,44]. Overall, our results show how an ROH-based method can be used to inform understanding of shifts in cultural marriage/mating practices.

As a second major finding, we observed that human background relatedness as measured by short ROH (4–8 cM) decreased markedly over time in many geographic transects, with a significant drop occurring during or shortly after the local "Neolithic Transition", the transition from a lifestyle of hunting and gathering to one of agriculture and settlement[45–47]. Assuming that early farmers had no increased individual mobility compared to foragers, which would agree with observations in present-day forager populations[48], the substantial decrease of short ROH evidences markedly increasing local population sizes. This finding adds support to the long-held hypothesis of local population sizes increasing following the Neolithic transition[45–47]. Previous analysis of ancient genomes of foragers and early farmers already identified several lines of genomic evidence for farmers having larger population sizes than earlier hunter–gatherers, such as decreasing genome-wide diversity[49,50], decreasing prevalence of ROH[11–14,18] and decreasing coalescent rates estimated from high-coverage genomes[27]. Our analysis adds a refined level of geographic and temporal resolution by analyzing an order of magnitude of more individuals (1785 ancient humans) and by organizing those individuals into several densely sampled time transects in different geographic regions.

For individuals from early Eurasian Steppe pastoralist groups, we observe an intermediate level of short ROH. These early cultures (e.g., the Yamnaya) have drawn much attention in archeological and ancient DNA studies to date, as archeological, linguistic, and genetic evidence suggest they played an important role in the origin of Indo-European languages and of several populations expansions[32,51–54]. The elevated rate of short ROH we observed provides evidence that many matings occurred within and among small, related groups. An alternative interpretation for the abundance of short ROH could be that burial sites (Kurgans) represent a biased sample of societal classes with more short ROH than the general populace[51]. However, as short ROH probes parental ancestry up to several dozen generations into the past, this signal would require reproductive isolation between societal strata maintained over many generations. Therefore, it is likely that at least part of the signal is due to Steppe populations having comparably low population densities or experienced recent bottlenecks.

Our analysis is limited by several caveats. Importantly, skeletal remains accessible by archeological means often do not constitute a random cross-section of past populations. While levels of background relatedness are expected to be similar within a mixing population, rates of close kin unions can vary substantially because of social structure; e.g., elite dynasties may practice close kin unions despite them being uncommon in the general population. Another limitation is the incomplete sampling of the current aDNA record and that for much of the world, we necessarily make inferences from small numbers and sparse sampling. Future work analyzing the rapidly growing ancient DNA record will help to resolve additional details of social and cultural factors operating at finer scales (e.g., leveraging more precise timings of shifts and more subtle shifts in ROH patterns). In particular, future studies focusing on specific localized questions will increasingly combine archeological and genetic evidence[16], in ways that will empower the use of the genetic evidence about the past provided by the methodology presented here.

In addition to denser sampling, there are several ways how our analysis can be improved upon by future work. Here we focused our analysis on long ROH (>20 cM) and short ROH (4–8 cM). While this dichotomy helped us to disentangle more clearly recent and distant parental relatedness, we expect that future work refining the downstream analysis of ROH will be able to extract more subtle signatures by looking across all ROH scales. Furthermore, we note that our application focused on a set of SNPs widely used for human ancient DNA (1240K SNPs). For whole-genome sequencing data (available for a subset of the data analyzed here), using all genome-wide variants would likely lower the requirements for coverage below the current limit of 400,000 of the 1240K SNPs covered at least once (corresponding to ca. 0.3× whole-genome sequencing coverage). Another improvement would be using a reference panel that includes ancient haplotypes. Currently, no long-range phased ancient haplotypes are available, but future work will likely produce such data.

One alternative approach to identify ROH in low coverage ancient genomes could be to use imputation followed by screening for stretches of homozygous markers using standard ROH detection methods. This was recently done for ancient individuals with >10× coverage[18]. Since imputation of genomes was reported to work well to a coverage similar to the low coverage cutoff used here [55,56ca. 0.5×] and most imputation methods are based on haplotype-copying methods related to the approach utilized here [the Li and Stephens model[22], we expect any such approach to perform similar to ours, after appropriate testing and calibration, as conducted for our method. We chose to develop a method utilizing several key advantages of pseudo-haploid data, which is more widely available and requires fewer assumptions about genotype quality, making subsequent analysis less prone to batch effects introduced by various isolation, sequencing, and genotyping protocols.

Identifying ROH can also be a starting point for other powerful applications: ROH consists of only a single haplotype (the main signal of our method), which is therefore perfectly phased, a prerequisite for powerful methods relying on haplotype copying[57] or tree reconstruction[26,58]. Moreover, long ROH could be used to estimate contamination and error rates, an important task in ancient DNA studies[20]. ROH lacks heterozygotes, allowing one to

identify heterozygous reads within ROH that must originate from contamination or genotyping error, similar to estimating contamination from the hemizygous X chromosomes in males[59]. Another promising future direction is the development of a method to identify long shared sequence blocks in ancient DNA not only within (ROH), but also between individuals, called identity-by-descent (IBD). Calling IBD between individuals would substantially increase power for measuring background relatedness since signals from every pair of individuals could be used. Moreover, a geographic IBD block signal is highly informative about patterns of recent migration[35,60–62]. Extending our method to similarly use haplotype information from a phased reference panel when detecting IBD could enable such analyses in low coverage ancients individuals.

Finally, the analysis of ROH has additional implications beyond human demography and kinship-based mating systems. In many plants and animal species, ROH is more prevalent (due to different mating systems, small population sizes, or domestication), and the study of ROH may be particularly interesting for understanding early plant and animal breeding, as actively controlled mating among domesticates would be expected to alter ROH[63]. For aDNA from extinct or endangered species, ROH can shed light on the extinction and inbreeding processes, as is observed for example in aDNA from high-coverage Neanderthal individuals[17,64–66], or modern DNA from Isle Royal wolves[67]. Finally, as ROH exposes rare deleterious recessive alleles[68], the temporal dynamics of ROH are relevant for understanding the evolutionary dynamics of deleterious variants and health outcomes[67,69–71]. We hope that the core ideas of our approach will inspire the analysis of low-coverage data from a wide range of natural populations.

## Methods

**Calling ROH in a global dataset**. To detect ROH, we developed a method, hapROH, which is based on an HMM with ROH and non-ROH states and uses a panel of reference haplotypes. The detailed method description and evaluation are provided in Supplementary Note 1, Supplementary Note 1.7, Supplementary Note 2.1 and (Supplementary Note 2.4). The software is publicly available at https://pypi.org/project/hapROH/. Fort the global data analysis we run hapROH (version 0.1a4) using the default parameter settings.

**Empirical dataset**. The global ancient DNA dataset we analyze originates from a curated dataset of published ancient DNA ("1240K", v42.4, released on March 1, 2020, https://reich.hms.harvard.edu). This release provides ancient DNA data in a pseudo-haploid format for 1.24 million SNPs (The 1240K SNP panel). This data includes whole genome as well as 1240K SNP capture data compiled from 92 primary publications which were processed starting from bam- or fastq-files using largely identical pipelines across datasets, only adjusting bioinformatics procedures when required by different data generation procedures. We added an additional 40 ancient Sardinian individuals in pseudo-haploid format from a recent publication[72] that had not yet been compiled into the global reference dataset.

We only analyzed previously generated, publicly available genetic data. For all data, we contacted the corresponding authors of each original study regarding our project and publication plan. We included in our final analysis the data from all studies for which we received a response confirming the use is consistent with the original permits. We filtered to individuals that contained PASS in the ASSESSMENT column of the meta-data table in order to remove individuals with possible contamination. For the remaining ancient individuals that had multiple genotypes listed, we kept the record with the highest coverage. Furthermore, we removed all Neanderthal and Denisovan individuals, as well as the individual tem003, for which initial analysis showed that it has all of chromosome 2 in ROH, but no other long ROH. Finally, we kept only individuals with at least 400,000 SNPs of the 1.24 million covered, the approximate cutoff above which our method can provide robust ROH inference (Fig. S1). For present-day data, we downloaded the Human Origins dataset with diploid genotype calls for ca. 550,000 autosomal SNPs[27], which are a subset of the 1240K SNPs.

We applied hapROH to the pseudo-haploid data for the 1785 ancient individuals and the diploid data for the 1941 modern individuals that pass our quality thresholds. We used all SNPs with available data for which the reference and the alternative allele matched the information in the reference panel, set the respective emission probabilities to values designed for these two types of data (Supplementary Note 1.3), and used the default parameters of hapROH that were optimized for the 1240K SNPs (Supplementary Note 1.8). For the haplotype

reference panel, we used the full global set of 5008 phased haplotypes of the 1000 Genomes Project dataset (Phase 3, release 20130502) accessible via http://ftp.1000genomes.ebi.ac.uk[24], filtered to biallelic markers and downsampled to SNPs in the 1240K SNP panel with bcftools (version 1.9). This standard human reference dataset is computationally (and in some cases trio-) phased and we kept the phasing as provided. Throughout, we used allele frequencies calculated from the diploid genotypes of the full reference panel when calculating the emission probabilities. We report the detailed ROH calls for all individuals in Supplemental Information 1.

**Annotation of subsistence strategy**. For each ancient individual, we annotated the primary subsistence strategy into standard broad categories of food production[73], using descriptions of the archeological sites and cultural affiliations. We used three main labels: We denoted (1) hunter–gatherer and horticulture lifestyles based on collecting wild plants, hunting, or fishing with the label "Forager"; (2) groups that practiced substantial amounts of sedentary farming (e.g., cereals and domesticates observed in the archeological record) as "Agricultural", and (3) groups with nomadic and semi-nomadic mobile lifestyles based on herding and breeding of domestic animals (e.g., cattle) as "Pastoralist". Groups that had intermediate and transitory lifestyles were annotated using the plausible dominant food economy of the associated archeological culture. To better resolve the transition to agricultural food production, we denoted early groups that practiced agriculture, but lack ceramics in the archeological record as "Aceramic Farmers". Individuals and groups for which the archeological record does not contain sufficient information to annotate a subsistence strategy were labeled as "Uncertain". We stress that archeological evidence is often sparse and assignments are frequently interpretations of various lines of evidence, therefore assessments might change with updates to the archeological record. Here, we tolerate some error, since we address questions regarding very broad temporal and geographic patterns, but we advise against using our subsistence assignments as a reference for questions on a finer scale.

**Detecting offspring of close relatives from ROH**. We screened all individuals for ROH longer than 20 cM to identify potential offspring of close relatives. Pairwise IBD > 20 cM, which translates to ROH in the offspring, is very unlikely to be a concatenation of multiple shorter IBD blocks[74]. Moreover, recombination quickly breaks up long ancestry segments of the genome, and thus most long ROH originates from co-ancestry within only a small number of generations back. Therefore, if the fraction of the genome in ROH longer than 20 cM in an individual is large, this provides strong evidence for a close relationship of its parents. We report individuals where the sum of all such ROH exceeds 50 cM as potential offspring of closely related parents (i.e., $sROH_{>20} > 50$). This cutoff is motivated by analytical calculations and simulations, see Supplementary Note 4 and Supplementary Note 5 for details. Briefly, this threshold detects a large fraction of close kin offspring (parents being a first cousin or closer) while also being insensitive to background relatedness unless a population has a very small size (<500).

**Gaussian process modeling of short ROH**. To visualize the trend of the abundance of ROH in the individuals in certain regions over time, while still conveying the levels of uncertainty due to varying sample sizes, we fit a Gaussian Process (GP) model[75] using the Python package `scikit-learn`[76]. As input, we used the square root of the $sROH_{[4,8]}$ statistic to stabilize its variance[77], since $sROH_{[4,8]}$ corresponds closely to count data, which can be approximated by a Poisson distribution. Furthermore, since we use this statistic as a proxy for background relatedness (which in turn proxies for local population size), we removed all individuals with $sROH_{>20}$ above 50 cM when fitting the GP model, to minimize the impact of putative offspring of close kin on this analysis (Fig. S13).

For the variance model of the GP, we used a standard squared-exponential covariance kernel summed with a residual white noise kernel. In preliminary analyses, we estimated all parameters of the model via maximum likelihood, but we found that these estimates appeared to over-fit the data for several time transects. Thus, we set custom length scales for the covariance kernel for each transect (1500 for all non-American populations and 2000 for American populations, because they had larger temporal sampling gaps) and only fit the two coefficients of the squared-exponential and white noise kernel. To visualize the final output, we estimated the variance of the predicted mean across a dense set of time points[75]. We estimated the uncertainty of the predicted mean and the uncertainty of each individual point and plotted both as 95% confidence interval bands (±1.96 standard deviations) on a dense grid.

**Analytical expectations of ROH**. To aid interpretation of ROH, we visualize expectations of sROH using formulas describing ROH of closely related parents in otherwise outbred populations and finite populations without substructure. We derive and state these formulas in a unified framework (Supplementary Note 4). We note that these formulas have been derived previously[78,79]. In addition, we verified these formulas by simulating ROH for these demographic scenarios and comparing expected sROH values to empirical averages (Supplementary Note 4 and Supplementary Note 5).

**Comparing ROH between groups**. To test significant differences in the distributions of the sROH statistics between two groups, we applied the Permutational multivariate analysis of variance method [PERMANOVA[80]], which calculates a pseudo-F statistic and assesses its significance via permutation tests. We used the `permanova` function implemented in the Python package skbio, and based the distance matrices on absolute differences of individual's $sROH_{[4,8]}$. For each test, we ran 99,999 permutations (minimal p-Value: $p = 10^{-5}$) and report two-sided p-Values. As with the GP modeling, we removed all individuals with $sROH_{>20}$ above 50 cM when comparing distributions of $sROH_{[4,8]}$ between groups.

**Reporting summary**. Further information on research design is available in the Nature Research Reporting Summary linked to this article.

## Data availability

No new DNA data were generated for this study. The ancient dataset and modern data [Human Origins[27]] we analyzed originate from the Allen Ancient DNA Resource (Version V42.4, available via https://reich.hms.harvard.edu), primary publications listed in Supplementary Data 1B. The raw reference panel data that we used (phased haplotypes from the 1000 Genomes dataset) is available at http://ftp.1000genomes.ebi.ac.uk/vol1/ftp/release/20130502/. The ancient and modern data we screened for ROH, as well as the processed reference panel we generated (down-sampled to biallelic SNPs at 1240k sites), are archived at https://doi.org/10.5281/zenodo.4992532. The source data underlying Fig. 2, Fig. 3, and Table 1, i.e., the ROH results on ancient DNA data, are provided in Supplementary Data 1A. Code that generates the data for each figure in the main text and Supplementary Information is listed in Supplementary Data 1C.

## Code availability

The Python package implementing the method is available at the Python Package Index (https://pypi.org/project/hapROH/) and can be installed using pip. The documentation provides example use cases as blueprints for custom applications. Code developed for simulating data, analysis, and data visualization is available at the GitHub repository https://github.com/hringbauer/hapROH. The version used for this work is archived at https://doi.org/10.5281/zenodo.4992416[81]. For data analysis we used Python (3.7.6) and the Python packages jupyterlab (2.1.2), scipy (1.3.1), pandas (1.1.4), numpy (1.19.4), and scikit-bio (0.5.6). We visualized results using matplotlib (3.1.1) and basemap (1.2.1).

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

## Acknowledgements

We thank the original study authors for sharing their data publicly, and David Reich and his lab, in particular Shop Mallick, for compiling and making publicly accessible a normalized pseudohaploid compilation of those data. We thank David Anthony and Alissa Mittnik for reviewing parts of the subsistence strategy annotations and for helpful discussions. We thank Arjun Biddanda, Shai Carmi, David Schloen, Lars Fehren-Schmitz, Montgomery Slatkin, and Mashaal Sohail for their comments on the paper. Funding for H.R. and J.N. was provided by NIH grant R01HG007089 and R01GM132383 to J.N.

## Author contributions

We annotate author contributions using the CRediT Taxonomy labels (https://casrai.org/credit/). Where multiple individuals serve in the same role, the degree of contribution is specified as "lead", "equal", or "support". Conceptualization (Design of study)—lead: H.R.; support: J.N. and M.S. Software—lead: H.R.; support: M.S. Formal analysis—H.R. Data curation—H.R.; support: J.N. Writing (original draft preparation)—lead: H.R.; support: J.N. and M.S. Writing (review and editing)—input from all authors. Supervision —equal: J.N. and M.S. Project administration—equal: J.N. and M.S. Funding acquisition —J.N.

## Funding

## Competing interests

The authors declare no competing interests.
