## [Peer Review File · Nature Communications]

REVIEWER COMMENTS

Reviewer #1 (Remarks to the Author):

The present manuscript by Ringbauer, Novembre & Steinrücken presents a method that leverages haplotype patterns in a reference panel to infer runs of homozygosity (ROH) in ancient DNA SNP capture data of low coverage. The method demonstrates that it is able to infer 4 centimorgan (cM) in individuals with coverage as low as 0.3x reliably.

This seems like a useful tool indeed for the ancient DNA field, but a few questions arise regarding the improvement of this method and novelty of results over previous approaches and studies. Firstly, a related manuscript by Ringbauer et al. 2020 that uses this method and cites the preprint version of this manuscript was recently published in *Current Biology* ([https://www.cell.com/current-biology/pdf/S0960-9822\(20\)31097-6.pdf](https://www.cell.com/current-biology/pdf/S0960-9822(20)31097-6.pdf)) and a question is then whether the full results should have been published together. Secondly, to which extent would the present method represent an improvement over a possible approach of coupling imputation of ancient genomes (Hui et al. 2020 *Scientific Reports*) with standard inferences of ROH using low-coverage genomes (Lipatov et al. 2015, bioRxiv; Cassidy et al. 2020, *Nature*)?

In addition, the authors find a decrease in short ROH following the Neolithic transition, noting "our results provide a new genetic line of evidence for the inference that local population sizes increased following the Neolithic transition", but this has been noted for some time in the ancient DNA literature, at first in two 2014 studies using estimates of conditional heterozygosity (Skoglund et al. 2014, *Science*), or genome wide patterns of TMRCA (Lazaridis et al. 2014). To which extent is this finding different from previously noted differences in N_e before and after the Neolithic transition?

Reviewer #2 (Remarks to the Author):

This manuscript entitled "Human parental relatedness through time revealed by runs of homozygosity in ancient DNA" by Ringbauer et al presents a new approach to detect runs of homozygosity (ROH) from ancient DNA (aDNA). The authors show that the approach can perform reliable inference with very low coverage data and hence analyze much more ancient samples than previous methods. Here 1,785 samples out of the 3,723 available (Reich Lab) when only 134 have been studied for ROH to date. With this increased sample size, the authors are able to study human DNAs at different points in time (up to 45k years ago) and in different geographical locations. They find that ancient human populations before the Neolithic transition (10k BP) had a lower prevalence of offspring from close kin unions, but showed higher background relatedness likely due to smaller population size.

The second part of the conclusion is in line with previous findings, but the first part was slightly unexpected. The authors do comment in the discussion that they could have missed some long ROH but even in the case where they missed 50% of them, ancient individuals would still show less inbreeding than modern-day individuals.

The newly proposed method hapROH is simply clever and rely on the fact that sequences in ROH are indeed identical DNA copies. It hence uses the pseudo-haplotype information from low coverage data (here human aDNA) in combination with a high coverage reference panel (here modern day 1000 Genomes individuals). Previous approaches all required diploid information. The authors evaluate the approach through simulations and real data from both ancient and modern DNA.

The manuscript is well written and clear. I do have however some questions and precisions on the methods.

1) One comment on vocabulary: I would tend to keep the term ROH for observational methods such as PLINK and use identical-by-descent (IBD) or homozygous-by-descent (HBD) segment for inference methods based on hidden Markov model (HMM). There seems to be a trend in recent literature for that (Ceballos et al 2018, Narasimham et al 2016) as opposed to previous publications, e.g. reviews Thompson 2013 (10.1534/genetics.112.148825), Browning 2012 (10.1146/annurev-genet-110711-155534). It is also more coherent with the use of IBD by the authors in the discussion.

2) There are other HMM approaches to inbreeding and HBD segment inference. Even if not directly compared, it would be interesting to comment on them, such as Viera et al 2016 (10.1093/bioinformatics/btw212), Leutenegger et al 2003 (10.1086/378207). Especially the specificities/advantages of the different hidden processes as they do not usually differ strongly on the emission probabilities.

3) Intuitively, I do not understand why using the diploid genotype likelihoods (=taking into account the 3 possible genotypes) is not doing better than pseudo-haploid genotype (=only one random draw of the 2 possible alleles). Here are a few checks I would be interested in seeing:

a. Within hapROH: diploid genotype likelihood vs. pseudo-haploid with the reference panel. In the current analysis of Ust Ishim man (supp 1.13, Fig S6): was there a comparison on low-coverage DNA (x0.3) of the pseudo-haploid genotype and the diploid genotype likelihood methods? I did not see it in Fig S6

b. In the comparison of hapROH (pseudo-haploid genotype & reference panel) with another HMM method bcftools/roh (diploid genotype likelihood but no reference panel)

i. This comparison is only done on simulated present day data (Supp2 TSI), could it be performed on real low coverage ancient data (such as down sampled Ust Ishim man) in order to have a more realistic temporal distance between the analyzed sample and the reference panel?

ii. Please specify for bcftools/roh how the allele frequencies are estimated: I am assuming that it is done on the 1000 Genomes (without TSI) that is also used as a haplotype reference for hapROH. But I could not find the information.

iii. The issue of fine tuning bcftools/roh (Supp 2, p20): as both approaches use very similar HMM (Supp 1.2), it should be possible to find equivalence in the transition rates. Intuitively going in/out of HBD should be the same, then within HBD would be treated differently.

iv. Results in Supp Tab 4 and Supp Fig 11 show false positive rate for bcftools/roh for diploid genotype data and not for hapROH, but the inverse for diploid genotype likelihood data (low read count data). Could you comment?

4) About hapROH hidden Markov model (HMM) parameters:

a. The parameters for the transition rates (suppl 1.8) have been chosen based on modern day Tuscany individuals with 1.24 million SNPs (1241K SNP panel) so that HBD segments have 5cM length and are composed of ~16 copy tracts of 0.3cM length (Suppl 1.8). How could that impact the analyses on ancient individuals? Impact of using only 400k-500k SNPs from 1241K SNP panel in the analyses?

b. It is also mentioned in the choice of posterior threshold (Supp 1.9 p8) that change in SNP set and reference panel will require changing the threshold. The differences in evaluated cut-of values seem very small.

c. From Suppl 1.10: "motivated by the observation that the vast majority of false positive ROH are

shorter than 2cM (Fig2), we only record ROH blocks > 2cM". Would changing the parameters to target 2cM HBD segment length rather than 5cM have help identify smaller HBD segments?

5) Long HBD segments:

a. The comment on p32 about long HBD segments not being the concatenation of multiple shorter IBD blocks: how does this fit with the transition rates of the HMM (multiple short copy tracts)?
b. Supplementary 2: "PLINK ... can break up long ROH, ... not observe when using bcftools/ROH or hapROH". Not a very fair comment as hapROH has been modified (section 1.10) to merge gaps between HBD segments. Does bcftools/ROH deal with long segments without the need for merging gaps?

6) Could repeating several times the process of generating the pseudo-haplotype genotype improve HBD inference? Since HBD regions will stay identical between iterations but non-HBD region would show variation.

7) Would adding some aDNA samples (e.g. samples >5x) to the modern-day reference panel improve HBD detection? In a similar way as a population-specific panel can be added to a reference panel when performing genotype imputation (e.g. Howie et al 2009, 10.1371/journal.pgen.1000529).

8) Should the use of a reference panel and the copying model of Li and Stephens be referred to as "linkage information" or "linkage disequilibrium information"? Several occurrences in the manuscript

9) Could you comment on any specificity of the distribution over the genome of the 1241K SNP panel, and the used subsets of 400k, 500k SNPs?

Minor points:

- Manuscript file

o Application (p6): the aDNA and modern data were "combined". Confusing term. It seems to imply that the same method (pseudo-haploid genotypes) and the same 400k SNPs were used for the analysis. But from methods: aDNA was analyzed pseudo-haploid data on 400k SNPs while for modern individuals, diploid genotypes and 500k SNPs were used.

o What is the error rate used for the analyses on the real data for hapROH: 3%?

o Figure 2 legend: upward triangle (panels C, D)? Color coding on maps vs. plots? "Horizontal line" is that the dashed line?

o Confusing referencing of the supplementary information that can be at the end of the main manuscript or in the supplementary file. Figure 4 & 5 exist in both.

- Supplementary file

o Fig 1: slightly confusing position of "Hardy Weinberg emissions" within the location of the "ROH segment"

o P3: there are indeed 3 models. Please adjust the sentence: "We implemented two emission models: ..."

o P6: Fig 2 C,D should be A, B?

o Fig 6 (Ust Ishim) & Fig 8 (Khomani 7): y-axis scale for the posterior probability (0 to 1?), cannot see blue dots below the posterior, order the posterior plots identically (e.g. pseudo-haploid top, diploid bottom). In both cases, the methods used were: pseudo-haploid genotypes and diploid genotypes (no read-count/genotype likelihood)? The text 1.13 /1.14 and figure legends can be confusing.

- o Fig 6 left figure: missing legend for color, horizontal and vertical lines. Add values of target coverage in addition to SNPs covered.
- o P19: SHRED-scale
- o P27: "as begin potential offspring" change to being
- o Fig 15: very hard to read the bottom labels on the right plots

Reviewer #3 (Remarks to the Author):

This is a review of "Human parental relatedness through time revealed 1 by runs of homozygosity in ancient DNA" by AUTHOR. I am voice typing this so my apologies for any sort of mistake.

The article describes a tool to infer runs of homozygosity in ancient samples. The authors claim it can perform well even down to 1X coverage. To illustrate what their tool can do, the user done several thousand ancient samples to show that runs of homozygosity were much more prevalent in the past.

The problem described by the authors is very real. Inferring runs of homozygosity is feasible for high-coverage data where the sites can confidently be called as heterozygous. This is not always the case for ancient samples and a tool that can solve this problem would be more than welcome. The authors are very candid about the limitations of their tool and it seems that it can only work for humans and more specifically Eurasians.

In general, the manuscript is very well-written and extensive. The authors have obviously put a lot of work into this manuscript. Although the conclusion that more ancient groups are somewhat more inbred is expected as they have less people to choose from. I have a few suggestions that could improve the manuscript. I have to say that I spent way more time in the supplemental as this is where the nitty-gritty details are.

I think my main questions regarding the description of the method, the simulations and the haplotype database.

-Description of the method

Supp page 2

An HMM has 2 statistical components, the emissions probability given a state and the probability of transition from a state to another. The authors have done really great work regarding the section about emission probabilities, it is very clear and easy to follow. However, the portion about the transition probabilities is a bit terser. On page 2 the paragraph "Following Li & Stephens, the copying states" .. is difficult to understand. I read it a few times and still was not able to make heads or tails of it. For an audience unfamiliar with the paper being cited, could you please elaborate and explain a bit more at length? Is a copying state an ROH state (e.g. sampling from haplotype #2 not #0 (hetero))?

Supp page 3

For the emissions probabilities, given an ROH state, can it flip from 0/0 (homo anc) to 0/1 (het anc/der) with probability epsilon?

For the same section about the emissions probabilities again: is epsilon a model parameter? a command-line parameter? How is epsilon determined?

If I miss this, I apologize it seems that the manuscript does not talk about runtime and memory requirements. This is an important aspect if people are to use this software.

-Simulated errors and testing

supp. page 6

Does the text mention that errors were added? How were they added, completely at random? If so, this might be an issue as C->T and G->A much more prevalent in ancient samples. what happens to the estimate when only C->T and G->A are added.

Supp page 10.

What is a 1% error rate for the genotype? Do you just flip a base?

Supp. page 19.

The paper talks about how they downsampled the data to mimic low coverage. What was the input data? Not BAM files I assume? Just genotypes? What is the same procedure done for Ust-Ishim? Or did you downsample a BAM file?

Was the method tested on WGS BAM files? I was not sure about whether it was but I think it should be done. And also the downsampling should be done at the BAM file level not simply using the method the authors describe to simulate downsampling in genotype calls.

In the ancient data, was there any sort of wetlab procedure to remove damage?

-Haplotype database

Main & Supplemental.

Both the main and the supplemental talk about a set of haploid genomes. However, how these are selected is insufficiently described. How were these haplotypes chosen? Is there a threshold on linkage between the SNPs?

In the same vein, how to deal with redundant haplotypes? If redundancy is not modeled, how to make sure that the allele frequencies are correct for the Hardy Weinberg equations for the non-ROH state?

Supp. page 15 Mentions that performance on Africans is poor. It seems to me that it under calls the ROH. Is this correct?

I thank the authors for being so candid about the performance of the method. However, this brings me to another concern. The paper claims that the method can work on anyone who descends from the out of Africa bottleneck. To justify this, the paper presents an analysis that was done on an ancient Asian genome using only European haplotypes. I had the following nagging concern. What happens if the panel of haplotypes is more drifted than the sample at hand? What happens if one tries to predict ROHs in a European sample using say a panel of Han Chinese haplotypes?

The reason why I ask is it possible that ancient hunter-gatherers in Eurasia had greater genetic diversity than what we can sample? Is it also possible that they had haplotypes that we have never been able to find? This could potentially mean that some of the estimates for more ancient samples are actually underestimated?

Miscellaneous

Supp page 4,

The last paragraph talks about the cutoff for T to call an ROH and says it is found in section 1.8, but I see it more in 1.9 on page 7. Could you double-check?

Supp. page 19

what is the SHRED scale?

Very minor point, I see a mix of British and American spelling: ex: page 1 supp “modeled” page 2 supp “modelled”. Please peruse the manuscript for consistency.

Supp page 4

Equation 1: please add some bounds for k, I imagine $1 \leq k \leq L$?

I managed to install the software. However, I have to say that I feel that there is insufficient details in the README. There should be a very simple section to say I have a set of BAM files what do I need to do? Instructions for bam files should ideally cover genotyping and LDL sampling. What happens if I have data in plink format, what are the next steps? How do I get the database of haplotypes? There is a link to a Dropbox but this is not great, I really platforms like <https://readthedocs.org/>

Point-by-point responses to reviewers

We restate the original comments, and then our responses (marked in **dark green**).

Reviewer #1 (Remarks to the Author):

The present manuscript by Ringbauer, Novembre & Steinrücken presents a method that leverages haplotype patterns in a reference panel to infer runs of homozygosity (ROH) in ancient DNA SNP capture data of low coverage.

The method demonstrates that it is able to infer 4 centimorgan (cM) in individuals with coverage as low as 0.3x reliably.

This seems like a useful tool indeed for the ancient DNA field, but a few questions arise regarding the improvement of this method and novelty of results over previous approaches and studies. Firstly, a related manuscript by Ringbauer et al. 2020 that uses this method and cites the preprint version of this manuscript was recently published in Current Biology ([https://www.cell.com/current-biology/pdf/S0960-9822\(20\)31097-6.pdf](https://www.cell.com/current-biology/pdf/S0960-9822(20)31097-6.pdf)) and a question is then whether the full results should have been published together.

Yes, a subset of us were authors on the Current Biology paper that used this method and cited our pre-print. We disclosed this manuscript applying the method to the editor in our initial submission letter, as well as another application of our method in the context of a manuscript about novel data from the Caribbean that was published in *Nature* on December 23rd 2020 (<https://doi.org/10.1038/s41586-020-03053-2>). We did not publish this manuscript jointly with those specific results for two main reasons: 1) We developed the present manuscript first with detailed attention to the methods and a global application; 2) Then some of the authors, in collaboration with new colleagues, applied the method to new data and with the purpose of investigating focused, regional-scale questions. While it would have been ideal to get this preprint to full publication first, we did not want to hold up applications of the method to newly generated data, and did not feel it was scientifically irresponsible as the methods and tests were fully available via pre-print for judgement, and we have been transparent in our cover letters regarding this situation. As the details of the method are not described elsewhere, nor is the large-scale global comparative application of the method, this manuscript will be the primary citation for both the method and our large-scale analysis of the pattern of human parental relatedness through time.

Secondly, to which extent would the presented method represent an improvement over a possible approach of coupling imputation of ancient genomes (Hui et al. 2020 Scientific Reports) with standard inferences of ROH using low-coverage genomes (Lipatov et al. 2015, bioRxiv; Cassidy et al. 2020, Nature)?

We acknowledge that imputation is another promising approach that could possibly extend the range of ancient DNA samples amenable to ROH analysis as well. We now included a paragraph about this alternative approach in the discussion. We discuss the advantages of

our method over this potential method, in particular, the ability to run hapROH on pseudo-haploid data which provides several key advantages for a broad global application:

[L443-L456]: “One alternative approach to identify ROH in low coverage ancient genomes could be to use imputation followed by screening for stretches of homozygous markers using standard ROH detection methods. This was recently done for ancient individuals down to $10^{\wedge}4$ coverage (Cassidy et al., 2020). Since imputation of genomes was reported to work well to a coverage similar to the low coverage cutoff used here (Hui et al., 2020; Rubinacci et al., 2021, ca. 0.5x) and most imputation methods are based on haplotype-copying methods related to the approach utilized here (the Li & Stephens model Li and Stephens, 2003), we expect any such approach to perform similar to ours, after appropriate testing and calibration, as conducted for our method. We chose to develop a method utilizing several key advantages of pseudo-haploid data, which is more widely available and requires fewer assumptions about genotype quality, making subsequent analysis less prone to batch effects introduced by various isolation, sequencing, and genotyping protocols.”

In addition, the authors find a decrease in short ROH following the Neolithic transition, noting "our results provide a new genetic line of evidence for the inference that local population sizes increased following the Neolithic transition", but this has been noted for some time in the ancient DNA literature, at first in two 2014 studies using estimates of conditional heterozygosity (Skoglund et al. 2014, Science), or genome wide patterns of TMRCA (Lazaridis et al. 2014). To which extent is this finding different from previously noted differences in N_e before and after the Neolithic transition?

Genetic evidence for increasing population sizes with the Neolithic transition had already been presented previously. We agree that our writing was not sufficiently clear about that -- we meant an “additional” line of genetic evidence, not the “first”. However, we stress that our results are novel in that the resolution is higher than in previous studies. We modified the paragraph of the discussion to more accurately reflect the relationship to previously published results, adding citations of relevant previous work:

[L385-L397]: “This finding adds support to the long-held hypothesis of local population sizes increasing following the Neolithic transition (Diamond and Bellwood, 2003; Ammerman and Cavalli-Sforza, 2014; Bacci, 2017). Previous analysis of ancient genomes of foragers and early farmers already identified several lines of genomic evidence for farmers having larger population sizes than earlier hunter gatherers, such as decreasing genome-wide diversity (Skoglund et al., 2014; Kousathanas et al., 2017), decreasing prevalence of ROH (Gamba et al., 2014; Jones et al., 2015; Broushaki et al., 2016; Sikora et al. 2017; Cassidy et al., 2020) and decreasing coalescent rates estimated from high-coverage genomes (Lazaridis et al., 2014). Our analysis adds a new level of geographic and temporal resolution by analyzing an order of magnitude more individuals (1,785 individuals) than hitherto amenable to ROH analysis and by organizing those individuals into several densely sampled time transects in different geographic regions.”

Reviewer #2 (Remarks to the Author):

This manuscript entitled “Human parental relatedness through time revealed by runs of homozygosity in ancient DNA” by Ringbauer et al presents a new approach to detect runs of homozygosity (ROH) from ancient DNA (aDNA). The authors show that the approach can perform reliable inference with very low coverage data and hence analyze much more ancient samples than previous methods. Here 1,785 samples out of the 3,723 available (Reich Lab) when only 134 have been studied for ROH to date.

With this increased sample size, the authors are able to study human DNAs at different points in time (up to 45k years ago) and in different geographical locations. They find that ancient human populations before the Neolithic transition (10k BP) had a lower prevalence of offspring from close kin unions, but showed higher background relatedness likely due to smaller population size.

The second part of the conclusion is in line with previous findings, but the first part was slightly unexpected. The authors do comment in the discussion that they could have missed some long ROH but even in the case where they missed 50% of them, ancient individuals would still show less inbreeding than modern-day individuals.

The newly proposed method hapROH is simply clever and rely on the fact that sequences in ROH are indeed identical DNA copies. It hence uses the pseudo-haplotype information from low coverage data (here human aDNA) in combination with a high coverage reference panel (here modern day 1000 Genomes individuals). Previous approaches all required diploid information. The authors evaluate the approach through simulations and real data from both ancient and modern DNA.

The manuscript is well written and clear. I do have however some questions and precisions on the methods.

1) One comment on vocabulary: I would tend to keep the term ROH for observational methods such as PLINK and use identical-by-descent (IBD) or homozygous-by-descent (HBD) segment for inference methods based on hidden Markov model (HMM). There seems to be a trend in recent literature for that (Ceballos et al 2018, Narasimham et al 2016) as opposed to previous publications, e.g. reviews Thompson 2013 (10.1534/genetics.112.148825), Browning 2012 (10.1146/annurev-genet-110711-155534). It is also more coherent with the use of IBD by the authors in the discussion.

Thank you for your suggestion. We do agree with the reasoning that the term HBD more closely reflects the use of ROH segments as effective IBD within an individual, effectively using recombination as a clock. However, since 1) recent publications (e.g. Narasimham et al 2016, and the comprehensive review of Ceballos et al 2018, and also the method “ROHan”, Renaud et al 2019) use the term ROH, 2) long ROH are effectively synonymous to HBD when inferring these segments, and 3) Our method infers segments (or “runs”) of homozygosity by design - we believe that introducing a term deviating from the majority of the recent literature would create confusion among users.

For further clarification, we now explicitly mention the alternative term “HBD” the first time “ROH” is defined in the main manuscript (L51-L53), and we reference to a new paragraph in the Supplement (L531-L544) that discusses the subtle difference between HBD and ROH.

In this new paragraph, we also refer to previous work which shows that in case of long ROH/HBD >4cM these definitions are effectively identical for practical purposes:

Supplement, L527-L540: “By design, our method identifies stretches of homozygous markers (ROH), whereas in this section we define a segment to be ended by any recombination break-point before coalescence, a definition which enables us to derive analytical approximations. Such stretches are sometimes called segments that are autozygous or homozygous by descent (HBD), as they are similar to definitions of identity-by-descent blocks between pairs of individuals (Ralph and Coop, 2013). Importantly, ROH and HBD segments are not equivalent, as a recent recombination event ending an HBD segment is difficult to detect if coalescent times on both sides of it are within the very recent past, resulting in segments of mostly homozygous markers on both sides of the recombination. This is akin to the “conflation” of IBD segments studied in (Chiang et al., 2016). However, for the blocks of the lengths we infer and discuss throughout this work (≈ 4 cM), for all but extreme levels of consanguinity, HBD and ROH can be considered identical for practical purposes, as the conflation effect decreases for long segments, and should be very rare for segments >4 cM based on experiments with IBD segments (Chiang et al., 2016).”

2) There are other HMM approaches to inbreeding and HBD segment inference. Even if not directly compared, it would be interesting to comment on them, such as Viera et al 2016 (10.1093/bioinformatics/btw212), Leutenegger et al 2003 (10.1086/378207). Especially the specificities/advantages of the different hidden processes as they do not usually differ strongly on the emission probabilities.

We thank the reviewer for pointing out this relevant HMM work. We tried to improve the references to relevant previous work that used HMMs to infer ROH (including these citations) and also to make our novel contributions clearer by extending the respective paragraph in the beginning of the discussion:

L331-L338: “Our algorithm follows a long line of previous work utilizing Hidden Markov Models (HMMs) to infer such segments (e.g. Leutenegger et al., 2003; Auton et al., 2009; Vieira et al., 2016; Narasimhan et al., 2016a). A key methodological advantage here is to use hidden states that, within an ROH segment, copy from a reference panel of haplotypes to take advantage of haplotype information. This new tool enabled us to screen aDNA data from 1,785 individuals for ROH, an order of magnitude more ancient individuals than hitherto amenable for such analysis.”

3) Intuitively, I do not understand why using the diploid genotype likelihoods (=taking into account the 3 possible genotypes) is not doing better than pseudo-haploid genotype (=only one random draw of the 2 possible alleles).

We agree that genotype likelihoods contain, in principle, more information than pseudo-haploid data. However, in the low coverage regime the gain is limited, as the additional analysis described in the next response (to “a.”) shows. The intuitive reason is that for low coverage, most loci will have only one read - thus the pseudo-haploid data already contains most of the information. Moreover, obtaining reliable genotype likelihoods for

ancient DNA is particularly challenging, which explains why the existing ancient DNA literature is dominated by pseudo-haploid data. We now describe this reasoning more clearly in the supplement:

Supplement, L123-L134: “However, obtaining appropriate genotype likelihoods for ancient DNA is a topic of much ongoing research (e.g. Prüfer, 2018), as accurate genotype likelihoods rely on non-trivial models (modeling reference bias, interdependence of reads, genotype errors) that in turn depend on the type of ancient DNA data generated (e.g. shotgun or capture data, UDG treatment, length of sequencing reads, single versus double stranded sequencing). Therefore, we decided to use pseudo-haploid data in our empirical application as such data requires fewer assumptions. While in principle perfect genotype likelihoods contain more information than pseudo-haploid data, based on simulated data we expect to see only small differences when analyzing low-coverage data (Fig. S5). The reason is that at low coverage the majority of sites are only covered by at most one read, and thus the pseudo-haploid genotype data contains most of the available information already.”

Here are a few checks I would be interested in seeing:

a. Within hapROH: diploid genotype likelihood vs. pseudo-haploid with the reference panel.

We generated such a comparison, comparing both emission models on the same simulated data, describing the new analysis in detail in Supplement 2.1 (page 11), and visualized the results in Figure S5 (page 13):

These results show that even on perfect “simulated under the model” genotype likelihoods, without the practical challenges of obtaining such likelihoods, there is relatively little gain to be made by using genotype likelihoods for low coverage data, at the tradeoff of more restrictive assumptions.

In the current analysis of Ust Ishim man (supp 1.13, Fig S6): was there a comparison on low-coverage DNA (x0.3) of the pseudo-haploid genotype and the diploid genotype likelihood methods? I did not see it in Fig S6

For the reasons mentioned in the two previous points, we developed and tested the method extensively for pseudo-haploid or diploid genotype data. Thus, our analysis of empirical data uses exclusively those two types, and therefore we focus our experiments on such data.

b. In the comparison of hapROH (pseudo-haploid genotype & reference panel) with another HMM method bcftools/roh (diploid genotype likelihood but no reference panel)

We replaced the genotype likelihood mode of hapROH in Fig. S12 (page 26) with a comparison to the pseudo-haploid mode, creating the suggested comparison.

i. This comparison is only done on simulated present day data (Supp2 TSI), could it be performed on real low coverage ancient data (such as down sampled Ust Ishim man) in order to have a more realistic temporal distance between the analyzed sample and the reference panel?

To produce ground truth ROH of known length in the way we did for the TSI required having several phased individuals so that we could create mosaic individuals. High quality phased data for ancient individuals from the same context do not exist, so we could not apply this comparison pipeline.

ii. Please specify for bcftools/roh how the allele frequencies are estimated: I am assuming that it is done on the 1000 Genomes (without TSI) that is also used as a haplotype reference for hapROH. But I could not find the information.

Throughout, we used allele frequencies calculated from the respective reference panel. We now describe that in the methods and supplement. For the specific case of the bcftools/roh application mentioned here, we indeed used allele frequencies from the reference panel. We now specify this in Supplement:

L484-L486:

“For hapROH and bcftools/ROH we used allele frequencies calculated from the global reference panel used by hapROH. We used the default settings of each method unless specified otherwise.”

iii. The issue of fine tuning bcftools/roh (Supp 2, p20): as both approaches use very similar HMM (Supp 1.2), it should be possible to find equivalence in the transition rates. Intuitively going in/out of HBD should be the same, then within HBD would be treated differently.

Indeed there is a principal equivalence as we note in the Supplement when describing hapROH (L58-L62). However the actual post-processing is different (and we believe our gap merging to be one of several major improvements) therefore the “best performance” parameters differ. Due to this technical difference, we decided to independently identify the best parameters. We added text in the Supplement:

L519-L523:

“In principle, there is an equivalence of bcftools/ROH and hapROH HMM transition rates into and out of ROH segments; however in practice this equivalence is challenging to calibrate because of how the post-processing differs. Therefore we explored the parameter space of both methods independently.”

iv. Results in Supp Tab 4 and Supp Fig 11 show false positive rate for bcftools/roh for diploid genotype data and not for hapROH, but the inverse for diploid genotype likelihood data (low read count data). Could you comment?

These differences occur in two different data regimes: one is for very low coverage and the other with diploid genotypes. While previously we had not described this behavior, we now explicitly mention in the Supplement that False Positive Rate decreases for hapROH with increasing coverage:

L514-L517:

“Notably, hapROH starts inferring false positive blocks of 2-4 cM when coverage drops below 0.5x, however the false positive rate remains negligible for blocks >4 cM (the target ROH length in our empirical analysis), and decreases for increasing coverage (Fig. S12).”

While false positive rate decreases with increasing coverage for hapROH, for bcftools/ROH the false positive rate 1-2 cM increases with increasing coverage (figure not shown in main text, as the behavior of bcftools/ROH in the 1-2 cM regime does not affect the main conclusions in the paper where we focus on segments >4cM):

4) About hapROH hidden Markov model (HMM) parameters:

a. The parameters for the transition rates (suppl 1.8) have been chosen based on modern day Tuscany individuals with 1.24 million SNPs (1241K SNP panel) so that HBD segments have 5cM length and are composed of ~16 copy tracts of 0.3cM length (Suppl 1.8).

How could that impact the analyses on ancient individuals?

We show that issues we might expect to arise in ancient DNA, i.e. low coverage, varying error rates and varying distance to the reference panel, introduce no practically relevant biases within our targeted application range when using the default parameters (Fig. S4, Fig. S6). Genealogical distance to the reference panel was also mimicked by analyzing a global sample of modern individuals, many of whom have no recent shared ancestry with the reference panel, and doing so we found no bias and high correlation of diploid ROH calls (which are essentially ground truth as ROH >4cM is robustly called with most existing methods, see Fig. S11) and pseudo-haploid calls across all modern populations, except some Sub-Saharan ones (Fig S11).

Impact of using only 400k-500k SNPs from 1241K SNP panel in the analyses?

The impact of varying target coverage is explored in Fig. S4 (between 0.1x-1.0x), and also by downsampling Ust Ishim (Fig. 1, Fig. S7).

b. It is also mentioned in the choice of posterior threshold (Supp 1.9 p8) that change in SNP set and reference panel will require changing the threshold. The differences in evaluated cut-of values seem very small.

The reason for this small range of presented values is that we first ran 20 replicates for a wider range of parameters (not shown in the initial submission) to identify a refined range, and then followed up with 80 additional replicates for the best parameter range - showing the values only for this range in the initial draft. We now added values from the wider range to share that information with the reader (Table S1) and modified the table caption accordingly.

c. From Suppl 1.10: “motivated by the observation that the vast majority of false positive ROH are shorter than 2cM (Fig2), we only record ROH blocks > 2cM”. Would changing the parameters to target 2cM HBD segment length rather than 5cM have help identify smaller HBD segments?

We could not identify a set of parameters that worked robustly for ROH of target length 2 cM. We now describe this explicitly in a new paragraph:

Supplement, L266-L276:

“Despite intensive testing, we could not identify any set of parameters where 2 cM ROH was called robustly simultaneously to an acceptable false positive rate for ROH of this length. We explored a wide range of transition rates (including parameters tuned to target shorter ROH), but could not identify any set of parameters which provided a substantial improvement for shorter blocks. In particular the many false positives that appear at ca. 2 cM for low coverage individuals (Fig. S4) persisted. We believe one reason is areas of low SNP density, whose effects become more prominent at this length scale (Fig. S18). We believe little can be done to achieve sufficient resolution at an acceptable false positive rate in that parameter

range when running a genome-wide analysis. However, a promising future approach beyond genome-wide screens for ROH could be focused screening on areas of high SNP density.“

5) Long HBD segments:

a. The comment on p32 about long HBD segments not being the concatenation of multiple shorter IBD blocks: how does this fit with the transition rates of the HMM (multiple short copy tracts)?

As we reference in Supplement, L535-L540, ineffective recombination events are rare and multiple ineffective recombinations events within a ROH block >4 cM are exceedingly rare. Moreover, the average copying tract length we specified was 0.3 cM. Thus there are too few ineffective recombination events to substantially affect the transition rate in that parameter regime.

b. Supplementary 2: “PLINK ... can break up long ROH, ... not observe when using bcftools/ROH or hapROH”. Not a very fair comment as hapROH has been modified (section 1.10) to merge gaps between HBD segments.

Our goal is to compare the methods available to a user and how they have been applied in the ancient DNA literature. We optimized hapROH parameters with gap merging as an integral part of the method, and this is made available via the default settings. The other two methods were not designed around gap merging.

We now make clearer that our description of this issue is intended as a note for PLINK users:

L492-L494: “This observation suggests that ROH analysis with PLINK would benefit from a post-processing step, merging short ROH gaps similar to the post-processing included by default in hapROH to remove spurious gaps.”

Does bcftools/ROH deal with long segments without the need for merging gaps?

Our results show that bcftools/ROH does not introduce spurious gaps in high quality present-day DNA or simulated data (see Supplement, L492). However we did not explore this in actual ancient DNA - as exploring and calibrating the genotype likelihoods for empirical ancient DNA necessary for applying bcftools/ROH is a complex challenge.

6) Could repeating several times the process of generating the pseudo-haplotype genotype improve HBD inference? Since HBD regions will stay identical between iterations but non-HBD regions would show variation.

That is an interesting idea, for high coverage individuals repeating the ROH inference for independently sampled genotype data would be a potential improvement. However, there are three reasons why we did not explore this idea further:

1) For low coverage individuals (<1x coverage) mostly only one read per site is available and thus resampling does not change the sampled allele - so there would be little signal for comparing across iterations.

2) The empirical data set we analyzed has the constraint of only providing pseudo-haploid data (in eigenstrat format), and such data cannot be resampled.

3) In a re-sampling based approach, runs of identity-over-iterations would emerge by chance in non-ROH regions and one would have to classify those runs as being from an underlying ROH region or not. This would take one in the direction of developing an HMM, and such an HMM would probably be strengthened by knowing about haplotype patterns in a reference panel. In the end we might expect a method of similar complexity as our current approach, but with the drawback of having little additional power as the coverage approaches only one read observed per site (i.e. reason 1 above).

7) Would adding some aDNA samples (e.g. samples >5x) to the modern-day reference panel improve HBD detection? In a similar way as a population-specific panel can be added to a reference panel when performing genotype imputation (e.g. Howie et al 2009, 10.1371/journal.pgen.1000529).

Yes, in principle it would help our method to add ancient haplotypes. One would need phased ancient haplotypes in the reference panel, but at the moment no such high quality phased ancient individuals are publicly available. We added this outlook to the discussion:

L440-L442:

“Another improvement would be using a reference panel that includes ancient haplotypes. Currently no long-range phased ancient haplotypes are available, but future work might produce such data, e.g. by directly phasing data of high-quality ancient parent-offspring trios.”

8) Should the use of a reference panel and the copying model of Li and Stephens be referred to as “linkage information” or “linkage disequilibrium information”? Several occurrences in the manuscript

Thank you for identifying that inconsistency. We chose to use “haplotype information from a phased reference panel” and now apply the term throughout the manuscript.

9) Could you comment on any specificity of the distribution over the genome of the 1241K SNP panel, and the used subsets of 400k, 500k SNPs?

We added a new section in the Supplement that describes the SNP set (Section 6, “Properties of SNP Set”), including a figure visualizing the density of 1240k SNPs along each chromosome (Fig. S18).

Throughout, we did not specify any subset of 1240K SNPs: In the empirical analysis all SNPs with at least one read were used (see Methods L529-L530). In downsampling experiments, SNPs were downsampled uniformly at random, and we describe that now more clearly:

Supplement, L329-L333: “When downsampling to create pseudo-haploid data, each locus was kept with target probability p , and set to missing otherwise. We note that in ancient DNA studies applying 1240K capture, some loci have a systematically higher chance to be

covered than others. However, our simple downsampling model should be a useful approximation as long as there are no population genetic biases affecting rates of missingness”

Minor points:

- Manuscript file

o Application (p6): the aDNA and modern data were “combined”. Confusing term. It seems to imply that the same method (pseudo-haploid genotypes) and the same 400k SNPs were used for the analysis. But from methods: aDNA was analyzed pseudo-haploid data on 400k SNPs while for modern individuals, diploid genotypes and 500k SNPs were used.

That is correct, modern individuals were analyzed on HO diploid data. We modified the description to make clearer that we included all available 1240K SNPs in each ancient individual, and only analyzed the ROH calls jointly, and now avoid the term “combined”:

L158-L164:

“Within this dataset, we inferred ROH longer than 4 cM using all available 1240K pseudo-haploid data in all ancient individuals and using diploid data for the HO SNPs in all modern individuals. After confirming that ROH calls on pseudo-haploid and diploid data in modern individuals correlate closely (Pearson correlation coefficient $r=0.925-0.988$, Supp. Fig. S8), we analyzed the inferred ROH in ancient and modern individuals jointly.”

o What is the error rate used for the analyses on the real data for hapROH: 3%?

We added a new paragraph into the section “Choice of Parameters and Posterior Threshold” (Supplement, 1.8)

Supplement, L228-L236: “The emission probabilities described above (Section 1.3) contain an error rate parameter. Throughout our analysis and for the determination and further evaluation of parameters, we set $e=0.01$. This error rate is a representative use case for our method: Error rates cover sequencing errors, ancient DNA damage, both typically not exceeding 1% (Briggs et al., 2007; Glenn, 2011) and contamination (mostly below 5%, Fig. S19, and not all contamination results in erroneous reads). Below, we explore how various rates of genotype error behave when using the default error rate in the HMM, showing that our goal to set an error rate that works robustly in a wide range of scenarios (from no error to error rates of a few percent) was achieved (Fig. S4).”

o Figure 2 legend: upward triangle (panels C, D)? Color coding on maps vs. plots?
“Horizontal line” is that the dashed line?

- 1) We now describe these upward triangles in the figure caption: “Individuals with value larger than the upper y-axis limit are indicated as upward triangles on top of the panels.”
- 2) The color coding is different. On the map we can only visualize the age, in the timeline we also convey the subsistence information, as the age is already given by the x-position.
- 3) We modified the term “horizontal line” to “horizontal dashed line”.

o Confusing referencing of the supplementary information that can be at the end of the main manuscript or in the supplementary file. Figure 4 & 5 exist in both.

We changed the labels and references to the two figures at the end of the main manuscript to “Extended Figure”. We will update them in the final version according to instructions from the editor.

- Supplementary file

o Fig 1: slightly confusing position of “Hardy Weinberg emissions” within the location of the “ROH segment”

We updated the figure to reduce the possibility of confusion (Fig. S1).

o P3: there are indeed 3 models. Please adjust the sentence: “We implemented two emission models: ...”

We clarified that two emission models were used on real data, while the simplistic genotype likelihood model is intended only for comparison on simulated data (for the reasons outlined above).

Supplement, L66:

“We implemented two emission models that we applied for analyzing empirical data.”

Supplement, L103-L104:

“We also implemented an experimental third emission model designed for read count data simulated under an idealized model”

o P6: Fig 2 C,D should be A, B?

We fixed that typo.

o Fig 6 (Ust Ishim) & Fig 8 (Khomani 7): y-axis scale for the posterior probability (0 to 1?), cannot see blue dots below the posterior, order the posterior plots identically (e.g. pseudo-haploid top, diploid bottom). In both cases, the methods used were: pseudo-haploid genotypes and diploid genotypes (no read-count/genotype likelihood)? The text 1.13 /1.14 and figure legends can be confusing.

We modified the figures (now Fig S7 and Fig S9) as suggested:

- 1) We added y axis labels for the posteriors.
- 2) We ordered the sub-figures in the same way.
- 3) We updated the writing within the two mentioned sections (previously 1.13/1.14) aiming to improve the text for clarity. Importantly, we removed text referring to the read count mode, which we did not show in the figures. We apologize for the confusion and hope that these sections read more clearly now.

o Fig 6 left figure: missing legend for color, horizontal and vertical lines. Add values of target coverage in addition to SNPs covered.

We modified the figure (now Fig S7) and its caption to more clearly explain the figure.

o P19: SHRED-scale

Thank you for identifying that typo. We changed the term to “PHRED scale”.

o P27: “as begin potential offspring” change to being

Thank you for identifying that typo.

o Fig 15: very hard to read the bottom labels on the right plots

We reworked the captions of this plot.

Reviewer #3 (Remarks to the Author):

This is a review of "Human parental relatedness through time revealed by runs of homozygosity in ancient DNA" by AUTHOR. I am voice typing this so my apologies for any sort of mistake.

The article describes a tool to infer runs of homozygosity in ancient samples. The authors claim it can perform well even down to 1X coverage. To illustrate what their tool can do, the user done several thousand ancient samples to show that runs of homozygosity were much more prevalent in the past.

The problem described by the authors is very real. Inferring runs of homozygosity is feasible for high-coverage data where the sites can confidently be called as heterozygous. This is not always the case for ancient samples and a tool that can solve this problem would be more than welcome. The authors are very candid about the limitations of their tool and it seems that it can only work for humans and more specifically Eurasians.

We appreciate the summary and make the minor note that with our current reference panel we show how the tool works for a wide range of global populations, apart from some Sub Saharan hunter gatherer groups (a point which we clarify in the revised manuscript).

In general, the manuscript is very well-written and extensive. The authors have obviously put a lot of work into this manuscript. Although the conclusion that more ancient groups are somewhat more inbred is expected as they have less people to choose from. I have a few suggestions that could improve the manuscript. I have to say that I spent way more time in the supplemental as this is where the nitty-gritty details are.

I think my main questions regarding the description of the method, the simulations and the haplotype database.

-Description of the method

Supp page 2

An HMM has 2 statistical components, the emissions probability given a state and the probability of transition from a state to another. The authors have done really great work regarding the section about emission probabilities, it is very clear and easy to follow.

However, the portion about the transition probabilities is a bit terser. On page 2 the paragraph "Following Li & Stephens [CITATION], the copying states" is difficult to understand. I read it a few times and still was not able to make heads or tails of it. For an audience unfamiliar with the paper being cited, could you please elaborate and explain a bit more at length?

We reworded the paragraph and added new explanations. We hope that this addition improves its readability:

Supplement, L48-L52: "As in the Li & Stephens copying model (Li and Stephens, 2003), every reference haplotype $i=1, \dots, n$ is equally likely to be copied from. Thus the transition

probabilities between these copying states (copying a homozygous genotype from the respective reference haplotype) are symmetric, and the transition rates to and from a copying state $i=1, \dots, n$ do not depend on the reference haplotype i . "

Is a copying state an ROH state (e.g. sampling from haplotype #2 not #0 (hetero))?

That is correct, each copying state $i=1, \dots, n$ is a ROH state. See also Fig. 1, Supp. Fig 1 and the following paragraph in the Supplement:

L30-L34: "In each of these copying states (denoted here as the ROH states), we model the copying as in the original Li & Stephens model (Li and Stephens, 2003), with one important modification: We assume that the genotype of the focal individual y is homozygous for the allele of the reference haplotype that it copies from."

Supp page 3

For the emissions probabilities, given an ROH state, can it flip from 0/0 (homo anc) to 0/1 (het anc/der) with probability epsilon?

In the diploid genotype and the experimental read count mode, the unobserved latent genotype can emit the two other possible genotypes with probability $e/2$ respectively.

Pseudo-haploid data can be either 0 (ancestral) or 1 (derived). In our model, we directly describe emission probabilities for those two observations, and thus a pseudo-haploid genotype can only flip from 0->1 and 1->0 (with equal probability e).

We added a description to the respective paragraphs:

Supplement, L77-L80: "We extend these genotype probabilities to model possibly erroneous genotypes: A genotype is homozygous for the copied allele with probability $1-e$ and it is flipped to one of the two other possible genotypes with probability $e/2$ respectively."

Supplement, L91-L93: "An observed pseudo-haploid genotype can have two states (ancestral or derived), and the emission probabilities for the two possible observations are given as follows."

For the same section about the emissions probabilities again: is epsilon a model parameter? a command-line parameter? How is epsilon determined?

Epsilon can be specified by the user (as parameter "e_rate" in the python function) - however we note that our testing and optimization was done for $e=0.01$. This is the default parameter chosen with typical ancient data in mind. We expect error rates on that order or smaller. We now edited the supplement to clarify our approach (see response to reviewer #2, Supplement, L228-L236).

If I miss this, I apologize it seems that the manuscript does not talk about runtime and memory requirements. This is an important aspect if people are to use this software.

We added a new section in the supplement describing memory and runtime requirements for typical ancient DNA applications, as well as the scaling with numbers of individuals and number of SNPs (Supplement 1.10: Runtime and Memory Requirements).

-Simulated errors and testing supp. page 6

Does the text mention that errors were added? How were they added, completely at random? If so, this might be an issue as C->T and G->A much more prevalent in ancient samples. what happens to the estimate when only C->T and G->A are added.

We added the following paragraph to clarify and explain why we believe our simple model of introducing errors allows us to sufficiently explore the range of applications to empirical data considered in the manuscript:

L334-L343:

“In our simulations, and also our modeling, we add genotype errors with equal probability to all sites, flipping the observed pseudo-haploid call to the other allele at random at a given rate. We note that in practice, errors are heterogeneous, e.g. C->T and G->A transitions are more prevalent in ancient DNA than transversions. The degree of heterogeneity depends on a plethora of technical details such as UDG treatment, the ancient DNA preservation, the position on each read and also what bioinformatic filtering has been applied. Instead of applying a complex error model, which will likely fail to capture the full complexity regardless, we chose to model and simulate data with a uniform error model. We assume that by using a slightly elevated uniform rate error for the whole set of SNPs, we can approximate the effects of unmodeled sources of error.”

Supp page 10.

What is a 1% error rate for the genotype? Do you just flip a base?

This is correct. In pseudo-haploid data, we change the observed allele to the other allele at this SNP with probability $e=0.01$. The updates described above (see comment above, L334-L343 in the Supplement) describe this now more clearly.

Supp. page 19.

The paper talks about how they downsampled the data to mimic low coverage. What was the input data? Not BAM files I assume? Just genotypes? What is the same procedure done for Ust-Ishim? Or did you downsample a BAM file?

We downsampled pseudo-haploid genotypes or read counts per SNP. Downsampling a BAM-file and then re-calling pseudo-haploid genotypes or read counts per SNP is effectively equivalent to this procedure. We now describe downsampling more explicitly (see Supplement, L329-L333, listed also under comment 9 of reviewer #2).

For Ust Ishim we downsampled read count data used in a previous publication, directly created from a BAM-file with a pile-up and typical additional ancient DNA quality control steps such as clipping 3 bp off the reads at each end and demultiplexing - see Marcus et al 2020 (which we cite). Again, this is effectively equivalent to downsampling the BAM-file.

We extended the description of the downsampling procedure:

Supplement, L415-L425: “We first analyzed the publicly available diploid genotype data (included in the empirical dataset we used for this study) with the diploid mode of our method, manually checking that the calls >4 cM are true gaps of heterozygous markers. We then use this ROH data as a baseline to compare to and analyzed read count data for the 1240K SNPs from Ust Ishim man ($40\times$ average read depth on the target) - using publicly available post-processed data from Marcus et al. (2020). We first down-sampled these reads to lower coverage ($0.2-40\times$), randomly selecting subsets of reads. We then created pseudo-haploid data for all SNPs covered (1,115,315 of the 1240K variants were covered) by choosing one remaining read at random for each SNP with at least one read covering it. We applied hapROH to this down-sampled pseudo-haploid data and called ROH with its pseudo-haploid emission mode.”

Was the method tested on WGS BAM files? I was not sure about whether it was but I think it should be done.

This method is implemented, tested and applied to pseudo-haploid data in eigenstrat-format - the format most widely used in most ancient DNA analysis and the format used in the compiled global data available from the Reich lab. To analyze WGS BAM-files, we recommend converting the BAM-file to an eigenstrat-file following standard pipelines widely used in ancient DNA analyses, specific to the data type at hand (depending on UDG treatments, aDNA preservation, single/double stranded library preparation). We added a point in the documentation (<https://pypi.org/project/hapROH/>) to clarify that we recommend such custom processing of the BAM-file.

And also the downsampling should be done at the BAM file level not simply using the method the authors describe to simulate downsampling in genotype calls.

As we explained above, downsampling pseudo-haploid data or read count data is equivalent to downsampling BAM files.

In the ancient data, was there any sort of wetlab procedure to remove damage?

Yes, as is typical for most ancient DNA datasets. We did not perform these procedures, but our dataset includes data treated with various wet lab procedures (e.g. UDG treatment). We updated the description of the compiled dataset in the Methods section:

L507-L512:

“Summarizing briefly, this data includes whole genome as well as 1240K SNP capture data compiled from 92 primary publications which were processed starting from bam- or fastq-files using largely identical pipelines across datasets, only adjusting bioinformatics procedures when required by different data generation procedures.”

-Haplotype database
Main & Supplemental.

Both the main and the supplemental talk about a set of haploid genomes. However, how these are selected is insufficiently described. How were these haplotypes chosen? Is there a threshold on linkage between the SNPs?

We used the default 1000G phased reference dataset, downloaded from the standard link that we cite. Phased haplotypes are provided, and we now describe more explicitly that we use this phasing without further modification:

L537-L541: “This standard human reference dataset is computationally (and in some cases trio-)phased and we kept the phasing as provided.”

In the same vein, how to deal with redundant haplotypes? If redundancy is not modeled, how to make sure that the allele frequencies are correct for the Hardy Weinberg equations for the non-ROH state?

We are not sure what is meant by redundant here. There will be shared haplotypes across individuals and perhaps that is what is meant by redundant. In such a case, the Li & Stephens copying model favors modeling the copying path through the longest shared haplotype between the target haplotype and the panel. With regards to the allele frequency estimation, if the concern is that there are related individuals in the panel, the 1000 Genomes sought to sample unrelated individuals and any background relatedness should be relatively minimal in its effect on allele frequency estimates. Such effects are typically ignored in studies of the 1000 Genomes data. Finally, our tests showed that cross-continental reference panels still work for various present-day individuals, showing that our method can tolerate modest inaccuracies in allele frequencies.

Supp. page 15 Mentions that performance on Africans is poor. It seems to me that it under calls the ROH. Is this correct?

This is correct. We write:

Supplement, L449-L451: “A notable exception are certain Sub Saharan populations, in particular South and East African hunter gatherers, for which a substantial fraction of long ROH are not identified in the haploid data (Tab. S8).”

I thank the authors for being so candid about the performance of the method. However, this brings me to another concern. The paper claims that the method can work on anyone who descends from the out of Africa bottleneck. To justify this, the paper presents an analysis that was done on an ancient Asian genome using only European haplotypes. I had the following nagging concern. What happens if the panel of haplotypes is more drifted than the sample at hand? What happens if one tries to predict ROHs in a European sample using say a panel of Han Chinese haplotypes?

We added the test suggested by the reviewer (East Asians references for European targets), and also added one additional test using East Asia+African references for Europeans targets (see Fig. S6). We indeed observe the effect anticipated by the reviewer, but we explain why this behavior of a single drifted reference panel is not worrisome for our empirical analysis that uses a global panel of reference haplotypes. The additional new experiment with the

East Asia+Africa panel is meant to show how the detrimental effects of having a more drifted, non-matching panel (East Asia for European) can be rescued by including diversity from an African panel. We think this comparison is relevant because in our empirical we use a global panel that includes African haplotypes.

Fig. S6

We describe this new experiments in a new paragraphs:

Supplement, L388-L403: “Second, we tested how two other sets of reference haplotypes work for mosaic individuals generated from Tuscany (TSI) individuals, again testing our method on pseudo-haploid data on 1240K SNPs. Notably, when using East Asian reference haplotypes (EAS, 1008 haplotypes), we find that our method has limited power to call ROH at 80% overlap - with power as low as 50% for some length categories. This reduction of power is not observed when using a similar number of European haplotypes as reference for East Asian mosaics (Fig. S6A, Tab. S2). One explanation for this asymmetry could be due to differential drift - many models infer a smaller effective size for East Asians than in Europeans after the shared out-of-Africa bottleneck (Keinan et al., 2007), plausibly causing a differential rate of loss of haplotype diversity. To address this, we included African reference haplotypes (AFR, 1322 haplotypes) in addition to the East Asian haplotypes to add additional haplotype diversity. Indeed, the power to identify ROH within TSI mosaics was restored (Fig. S6B, Tab. S2). This observation indicates the benefits of using a global reference panel (as used in our empirical application).”

The reason why I ask is it possible that ancient hunter-gatherers in Eurasia had greater genetic diversity than what we can sample? Is it also possible that they had haplotypes that we have never been able to find? This could potentially mean that some of the estimates for more ancient samples are actually underestimated?

Yes - this does appear to be a problem as shown in the EAS/EUR experiment described in the previous response. That said, we believe the simulations of the EAS+AFR panel in a EUR test set more closely mimic the situation in our empirical analysis. By including the more haplotype rich sample from prior to the bottleneck, the effects of using a more drifted panel are compensated.

For further support that ancient hunter-gatherers are amenable to our analysis, consider our application to Ust Ishim man (e.g. Fig. 1) - with 45k years the oldest Eurasian HG sampled to date. From his high coverage data (ca. 40x) we have very accurate estimates of his

(sometimes long) ROH - and downsampling shows that pseudo-haploid data introduces no substantial bias, even at 0.3x coverage (see Fig. S7).

Miscellaneous

Supp page 4,

The last paragraph talks about the cutoff for T to call an ROH and says it is found in section 1.8, but I see it more in 1.9 on page 7. Could you double-check?

Thank you for identifying that erroneous reference. We now merged the two sections into a new section 1.8 “Choice of Parameters and Posterior Threshold” as they thematically fit together.

Supp. page 19

what is the SHRED scale?

We fixed it to the correct term “PHRED scale”.

Very minor point, I see a mix of British and American spelling: ex: page 1 supp “modeled” page 2 supp “modelled”. Please peruse the manuscript for consistency.

Thank you for spotting this inconsistency. We checked for several common British/American spelling differences and updated the manuscript to American English spellings.

Supp page 4

Equation 1: please add some bounds for k, I imagine $1 \leq k \leq L$?

We updated Equation 1 and also Equation 2 as suggested. As k enumerates the hidden states of the HMM, it runs from $i=0,1,\dots,n$ as our model has one background state and n copying states.

I managed to install the software. However, I have to say that I feel that there is insufficient details in the README. There should be a very simple section to say I have a set of BAM files what do I need to do? Instructions for bam files should ideally cover genotyping and LDL sampling. What happens if I have data in plink format, what are the next steps? How do I get the database of haplotypes? There is a link to a Dropbox but this is not great, I really platforms like <https://readthedocs.org/>

We prepared an improved documentation at the official Python package website <https://pypi.org/project/hapROH> that includes some of these questions as a FAQ. We also refer to our vignettes (to which the documentation refers) that walk through how to use the method on pseudo-haploid eigenstrat data (the most widely used format in human ancient DNA). We link to resources to how to produce such files and how to convert from other file formats to it. Generally, we have already received positive feedback from users who successfully used our vignettes to analyze their newly generated aDNA data, and we are committed to improving the documentation in response to user questions.

REVIEWERS' COMMENTS

Reviewer #1 (Remarks to the Author):

I am pleased with the responses to my specific comments and have no further comments to raise.

Reviewer #2 (Remarks to the Author):

The authors have adequately addressed my comments